# Progressive Cramming: Reliable Token Compression and What It Reveals

**Dmitrii Tarasov** [1 2]   **Timofei Lashukov** [2]   **Elizaveta Goncharova** [1 2]   **Andrey Kuznetsov** [1 3]

## Abstract

Token cramming compresses sequences into learned embeddings with near-perfect reconstruction, but fixed token budgets and 99% accuracy thresholds leave it unclear whether residual errors reflect optimization failures or fundamental limits. We introduce progressive cramming, which grows the target prefix token-by-token, stopping only when reconstruction is no longer achievable within a fixed optimization budget. Progressive trajectories occupy low-dimensional structure in embedding space. Prepending a crammed embedding causes a moderate but consistent accuracy drop on multiple-choice benchmarks even with the original prefix in context, and collapses capability almost entirely under generative evaluation. Causal attention-knockout interventions trace this degradation to the embedding's interactions in the model's early layers. These results position progressive cramming as a tool for studying compression limits and show that perfect reconstruction—achievable through brittle steering rather than transferable semantics—is insufficient for meaningful compression.

## 1. Introduction

How much information can a single embedding encode? Recent work on "cramming" (Kuratov et al., 2025) probed this question by optimizing embeddings to reconstruct token sequences through autoregressive decoding. The results were striking: a single input embedding can encode up to 1568 tokens with near-perfect reconstruction, suggesting substantial latent capacity in transformer representations.

But *how* do transformers achieve this? The original work focused on capacity limits – how many tokens fit – rather than

[1] FusionBrain Lab, AXXX, Moscow, Russia [2] HSE University, Moscow, Russia [3] Innopolis University, Innopolis, Russia. Correspondence to: Dmitrii Tarasov <dtarasov@hse.ru>.

*Proceedings of the 43rd International Conference on Machine Learning*, Seoul, South Korea. PMLR 306, 2026. Copyright 2026 by the author(s).

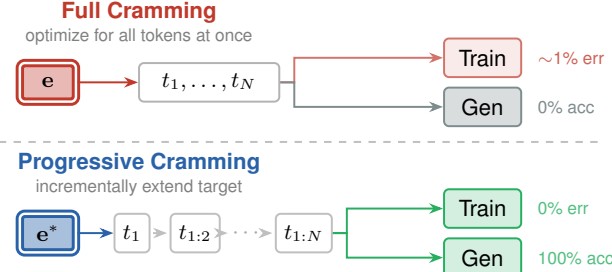

*Figure 1.* Full cramming achieves ∼1% training error with teacher forcing, but errors cascade to 0% generation accuracy. Progressive cramming establishes a precise boundary where compression either succeeds completely or fails clearly.

the mechanism enabling reconstruction. This leaves open a fundamental question: does cramming discover dense semantic representations that the model can interpret, or does it exploit some other property of transformer computation? Understanding this mechanism matters beyond cramming: it can shed light on how transformers use information stored in embedding space.

We first address methodological limitations that complicate controlled study. Prior work used fixed token budgets and 99% accuracy thresholds with teacher forcing. But the last 1% of errors may lead to catastrophic failures in autoregressive generation without teacher forcing. We introduce *progressive cramming*: sequentially adding tokens until perfect reconstruction, with a low-dimensional projection that stabilizes optimization. This achieves reliable 100% accuracy and enables precise measurement of where compression fails. Figure 1 shows comparison of full and progressive cramming frameworks.

With these tools, we investigate the cramming mechanism. Analysis of optimization trajectories shows they occupy low-dimensional manifolds: 30-100 PCA components explain 99% of variance in 2048-4096 dimensional embedding spaces. More critically, attention-pattern analysis shows that the compression embedding becomes a strong attention sink, capturing 20–60% of attention mass in intermediate layers. This concentration coincides with a breakdown of downstream language-modeling capabilities rather than benign attention focusing; our causal analysis (Section 7) further

shows it is a *symptom* of that breakdown rather than its cause. This concentration persists across sequence lengths – it appears whether cramming 32 or 512 tokens – consistent with compression engaging relatively fixed circuits rather than scaling information density.

This mechanism has consequences. If cramming worked through semantic encoding, compressed prefixes should preserve downstream capabilities. Evaluation on HellaSwag and ARC shows the opposite: accuracy consistently drops below the uncompressed baseline even when the original prefix remains in context, and under generative evaluation it collapses almost entirely. Together with the attention results, this suggests that cramming can achieve reconstruction via brittle steering (e.g., attention capture) rather than by encoding information the model can flexibly use.

These findings reframe what cramming reveals about transformers. The high capacity demonstrated by prior work reflects not dense semantic encoding but the ease of steering the model through its attention mechanisms. This distinction matters for future work on learned compression: methods that override computation can achieve perfect reconstruction while encoding nothing transferable.

Our main contributions are as follows:

- We introduce progressive cramming, which grows the target prefix token-by-token and targets 100% reconstruction, enabling a sharp success/failure boundary beyond fixed-budget, 99% protocols.

- We show that progressive optimization trajectories $\{\mathbf{e}^{(k)}\}$ are low-dimensional: across model families, 30–100 PCA components explain 99% of per-sample trajectory variance in 2048–4096 dimensional embedding spaces.

- We demonstrate that optimizing for reconstruction does not imply preserved downstream utility: on likelihood-based multiple-choice evaluation (HellaSwag, ARC-Easy), prepending a crammed compression embedding causes a moderate but consistent accuracy drop across model families even when the original prefix remains in context, and under generative evaluation it collapses capability almost entirely.

- Using *attention knockout* we causally localize this degradation to the compression embedding's interactions in the *early* layers: masking them restores downstream capability, whereas the late layers that carry the most attention mass have little causal effect. The embedding's attention concentration is therefore a *symptom* of cramming rather than the cause of capability loss.

- We show that cramming capacity is bounded by the frozen reconstructor: truncating pretrained models to

their first $N$ layers (then repairing the cut with a short finetune) reveals that compression capacity rises monotonically with both retained depth and model width.

- We publicly release our code and the set of progressive-cramming optimization trajectories.[1]

## 2. Related Work

### 2.1. Token Cramming

Our work is built upon token cramming (Kuratov et al., 2025), where per-sample optimization compresses up to 1,568 tokens into a single Llama-3.1-8B input embedding ($\sim 1500\times$ compression), far beyond typical encoder-based lossless ratios ($\leq 10\times$). Crucially, they show that capacity is better characterized by cross-entropy reduction than raw token count, suggesting the model leverages internal knowledge to complement stored information.

Extending this line of inquiry, Mezentsev & Oseledets (2025) show that frozen LLMs can generate hundreds of accurate tokens non-autoregressively from just two learned "proto-token" embeddings, and that valid compression embeddings form connected local regions in embedding space.

### 2.2. Context Compression and Distillation

A parallel line of research compresses context to reduce inference cost. Unlike token cramming, these methods target *semantic compression*—preserving task-relevant information for downstream use rather than *exact reconstruction* of the token sequence.

Zhang et al. (2025) propose Activation Beacon, which compresses per-layer keys and values into special beacon tokens, enabling flexible compression ratios and progressive chunk-wise processing up to 400K tokens while preserving short-context ability.

Chevalier et al. (2023) propose AutoCompressors, which learn unsupervised summary vectors that act as soft prompts, letting the model condition on compressed representations of earlier segments as it reads a document sequentially.

The In-Context Autoencoder (ICAE) (Ge et al., 2024) uses a LoRA-adapted encoder to compress contexts into memory slots that the target LLM conditions on, achieving $4\times$ compression while retaining prompt-following.

These methods target semantic compression for downstream tasks, whereas we study near-lossless reconstruction as a cleaner probe of embedding-based information storage.

---

[1] https://github.com/FusionBrainLab/progressive_cramming

## 2.3. Soft Prompts and Prefix Tuning

The optimization of continuous embeddings as model inputs originated in parameter-efficient fine-tuning. Li & Liang (2021) prepend learnable "virtual token" vectors that subsequent tokens attend to, matching full fine-tuning on generation tasks while optimizing only $0.1\%$ of parameters with the model frozen. Lester et al. (2021) simplify this to soft prompts at the input layer alone, and show they close the gap with full model tuning as scale grows—evidence that larger models develop embedding spaces rich enough for task-specific conditioning through input manipulation alone.

These methods establish that frozen models can be steered through learned embeddings without modifying parameters. Token cramming is an extreme case: the embedding must encode the full content of an arbitrary sequence rather than a task instruction. We study how it interacts with the model's attention and activation patterns as that demand grows.

## 3. Background: Token Cramming

### 3.1. Problem Formulation

In the cramming task, a single embedding is optimized to encode as many tokens as possible through autoregressive reconstruction. The language model weights remain frozen throughout optimization, and each sequence receives a newly initialized compression embedding.

Let $\mathcal{M}$ be an autoregressive language model with vocabulary $\mathcal{V}$ and embedding dimension $d$. Given a target sequence $\mathbf{x} = (x_1, \ldots, x_n)$ where $x_i \in \mathcal{V}$, the cramming objective finds an embedding $\mathbf{e} \in \mathbb{R}^d$ such that:

$$\mathbf{e}^* = \arg \min_{\mathbf{e}} \mathcal{L}_{\text{cram}}(\mathbf{e}; \mathbf{x}, \mathcal{M}) \tag{1}$$

where the cramming loss is the cross-entropy over the target sequence:

$$\mathcal{L}_{\text{cram}}(\mathbf{e}; \mathbf{x}, \mathcal{M}) = -\sum_{i=1}^{n} \log p_{\mathcal{M}}(x_i \mid \mathbf{e}, x_1, \ldots, x_{i-1}) \tag{2}$$

### 3.2. Limitations of Prior Work

Prior work on prompt compression and reconstruction typically evaluates success using a fixed accuracy threshold (e.g., 99% token-level reconstruction under teacher forcing), often in conjunction with a fixed token budget. Our results indicate that this evaluation protocol substantially underestimates critical failure modes that emerge during autoregressive generation.

**The "last 1%" problem.** While a 99% reconstruction accuracy suggests near-perfect training performance, the remaining errors are not uniformly distributed nor benign.

*Table 1.* Reconstruction brittleness on PG19. Despite high teacher-forcing convergence (TF conv.), greedy decoding from the compression embedding collapses (Greedy $\approx 0$) because the residual $\sim 1\%$ error is concentrated at the first two token positions (mismatch rate at indices 0/1). Full per-variant breakdown in Table 20.

| Model | TF conv. | Greedy | Mism. @0/1 (%) |
|---|---|---|---|
| Llama-3.2-1B | 99.0% | 0.2% | 86 / 100 |
| Llama-3.2-3B | 99.0% | 0.2% | 100 / 100 |
| Llama-3.1-8B | 99.1% | 0.4% | 100 / 100 |

As shown in Table 1, reconstruction mismatches are almost entirely concentrated at the earliest token positions, specifically at indices 0 and 1. Even when mean teacher-forcing convergence is high (95–99%), the probability of an incorrect prediction at the first or second token during inference approaches 100% across most model sizes and training variants.

**Early-token errors induce autoregressive collapse.** Errors at the earliest positions have a disproportionate impact under greedy decoding. Once the autoregressive process deviates at the first or second token, all subsequent tokens are generated conditioned on an incorrect prefix, leading to rapid and irreversible divergence. Consistently across all evaluated models, we observe that mean greedy convergence collapses to near zero (at most $\sim 4\%$) whenever compression training fails to achieve perfect reconstruction. In practice, a single early mismatch nullifies the utility of the remaining correctly reconstructed tokens, despite high aggregate training accuracy.

**Fixed-budget evaluation obscures brittleness.** Finally, evaluation under a fixed token budget further masks this brittleness. Methods with similar final convergence values may exhibit qualitatively different failure modes, depending on whether reconstruction errors occur early or late in the sequence. As demonstrated in Table 1, high average token-level accuracy does not guarantee usable autoregressive behavior. Instead, perfect reconstruction of the earliest tokens emerges as a necessary condition for reliable decoding from compressed representations.

## 4. Method: Progressive Cramming

### 4.1. Progressive Token Addition

Rather than optimizing a compression embedding for a fixed-length target, we progressively extend the target prefix and warm-start optimization:

1. Initialize with $\mathbf{x}^{(1)} = (x_1)$ and optimize $\mathbf{e}^{(1)} \in \mathbb{R}^d$ until perfect reconstruction.

2. Extend to $\mathbf{x}^{(k)} = (x_1, \ldots, x_k)$, initialize $\mathbf{e}^{(k)} \leftarrow$

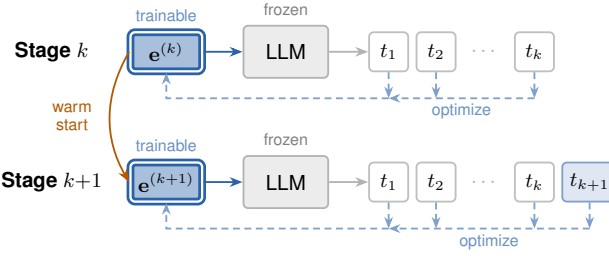

*Figure 2.* Progressive cramming adds target tokens sequentially. At stage $k$, we optimize the compression embedding to achieve 100% reconstruction of the prefix $(x_1, \ldots, x_k)$, then extend to $k+1$ and warm-start from the previous solution.

$\mathbf{e}^{(k-1)}$, and continue optimizing for the longer prefix.

3. Repeat until perfect reconstruction is no longer achievable within the optimization budget.

This process is visualized on Figure 2.

Progressive cramming produces a sequence of embeddings $\{\mathbf{e}^{(k)}\}_{k=1}^n$, where $\mathbf{e}^{(k)}$ is the optimized compression embedding for the length-$k$ prefix. This lets us quantify how far the optimizer moves in embedding space as new tokens are added. We define the (stage-wise) trajectory length as the sum of Euclidean distances between consecutive stages:

$$L_{\text{traj}} = \sum_{k=1}^{n-1} \left\| \mathbf{e}^{(k+1)} - \mathbf{e}^{(k)} \right\|_2 \tag{3}$$

### 4.2. Low-Dimensional Projection

We observed that optimization trajectories occupy low-dimensional subspaces (Section 5.1). This motivates constraining optimization to a learned projection:

$$\mathbf{e} = \mathbf{W}\mathbf{z} + \mathbf{b} \tag{4}$$

where $\mathbf{z} \in \mathbb{R}^k$ with $k \ll d$, $\mathbf{W} \in \mathbb{R}^{d \times k}$, and $\mathbf{b} \in \mathbb{R}^d$. For each sample, we randomly initialize $(\mathbf{W}, \mathbf{b}, \mathbf{z})$ and optimize them jointly, which restricts the compression embedding to a rank-$k$ affine subspace. While $(\mathbf{W}, \mathbf{b})$ are fit per sample by default, the same parameterization also admits a *shared* projection—a single $(\mathbf{W}, \mathbf{b})$ trained jointly across a corpus, with only the per-sample coefficients $\mathbf{z}$ left free—which lets us ask whether the low-rank subspace of valid solutions is corpus-universal or sample-specific.

After optimization, the effective embedding $\mathbf{e}$ can be materialized once and reused; the projection primarily changes the optimization geometry by introducing optimizer state (e.g., momentum) in the low-dimensional coordinates.

We additionally explore an activation-alignment regularizer that pulls the crammed hidden states toward their uncompressed counterparts; because it helps only some model families, we relegate it to Appendix B.5.

### 4.3. Information Gain

As a primary cramming metric we use information gain (Kuratov et al., 2025), defined as the reduction in total cross-entropy (in bits) on the target sequence when conditioning on the learned compression embedding. Let $H_{LM}$ be the sum of per-token cross-entropies for the original token sequence, and let $H_{\text{comp}+LM}$ be the corresponding quantity when the model is conditioned on the compression embedding.

$$C_H = H_{LM} - H_{\text{comp}+LM} \tag{5}$$

Kuratov et al. (2025) show that cramming capacity is better characterized by information gain than by raw token count, and that this quantity is relatively stable across datasets.

## 5. Experiments

We evaluate progressive cramming on PG19 (Rae et al., 2020) across four model families: Pythia (Biderman et al., 2023), Llama 3 (Dubey et al., 2024), SmolLM2 (Allal et al., 2025), and Gemma 3 (Team et al., 2025). To confirm that our findings are not specific to a single corpus, we replicate the main results on a second dataset (Fanfics, following Kuratov et al. (2025)) in Appendix D.4, where the same patterns hold—low-dimensional projection increases both compressed-token count and information gain, and compression trajectories remain low-dimensional.

Table 2 compares progressive cramming (targeting 100% reconstruction) to "full" cramming baselines that use fixed token budgets and may terminate below perfect accuracy. Because autoregressive decoding is brittle to early errors (Section 3.2), we interpret this comparison primarily through the lens of *perfect* reconstruction: progressive cramming trades a fixed-budget constraint for a fixed-accuracy constraint. In the default setting, progressive cramming sightly underperforms full cramming in reconstructed length. Crucially, however, progressive cramming guarantees perfect reconstruction under autoregressive generation, whereas full cramming, does not.

### 5.1. Optimization Trajectories

Progressive cramming enables tracking the optimization path through embedding space. We record the sequence of optimized embeddings $\{\mathbf{e}^{(k)}\}_{k=1}^n$ and analyze its geometry. For each sample, we run PCA on the stage embeddings $\{\mathbf{e}^{(k)}\}_{k=1}^n$ across $k$ (using standard PCA mean-

*Table 2.* Full vs. progressive cramming on PG19 (mean ± std over samples). "Full" cramming (Kuratov et al., 2025) uses a fixed token budget and may stop below perfect Teacher Forcing (TF) reconstruction; progressive cramming fixes reconstruction at 100% and guarantees perfect reconstruction under autoregressive greedy generation.

| Type | Tokens | Accuracy (%) | |
|---|---|---|---|
| | | TF | Greedy |
| **Llama-3.1-8B** | | | |
| Full | 1568 | 99.96 ± 0.04 | 40.44 ± 48.63 |
| Prog. | 1438 ± 380 | 100.00 | 100.00 |
| **Pythia1.4b** | | | |
| Full | 512 | 99.71 ± 0.65 | 44.19 ± 44.82 |
| Prog. | 430 ± 65 | 100.00 | 100.00 |

centering across stages) and define "PCA 99%" as the minimum number of components whose cumulative explained-variance ratio reaches 99%. We report this quantity averaged over samples. Across model families, trajectories are well-approximated by low-dimensional structure: a modest number of principal components explains 99% of the trajectory variance, even when hundreds (or thousands) of tokens are reconstructed (Table 3). We emphasize that this is a property of the optimization *path*, not of the underlying solution set: equally-good solutions reached from the same initialization by different learning rates are far apart and nearly orthogonal, so the set of valid compression embeddings is itself wide and high-dimensional, and the low-dimensional trajectory is one thin slice of it. The *shape* of this path is punctuated rather than smooth: progressive cramming proceeds as a "dwell-and-leap" walk that alternates runs of cheap tokens with occasional expensive ones, at every model scale.

Notably, Gemma 3 achieves lower information gain at shorter reconstructed lengths than Llama and Pythia, consistent with architectural or training-time constraints such as logit softcapping (Team et al., 2024).

Figure 3 visualizes a single trajectory projected onto the first two principal components. As the reconstructed prefix grows, the region of embeddings that attains near-perfect teacher-forced reconstruction contracts, suggesting that the optimization landscape becomes increasingly constrained at longer lengths.

Figure 4 shows that for Llama-3.1-8B, the number of components required to explain 99% variance grows sublinearly with prefix length and follows a slow (approximately logarithmic) trend across learning rates.

### 5.2. Low-Dimensional Projection

Low-dimensional projection (Section 4.2) changes the optimization geometry by restricting the learned embedding

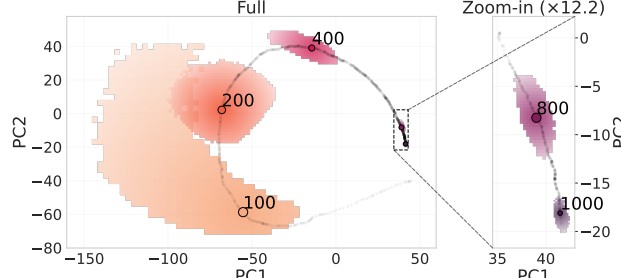

*Figure 3.* Progressive cramming trajectory for a length-1000 sequence on Llama3-8B, projected onto the first two PCA components. Each black point is the optimized compression embedding for a given prefix length. For prefix lengths {100, 200, 400, 800, 1000}, we additionally visualize the local accuracy landscape in this 2D plane (color saturation indicates higher reconstruction accuracy, capped at 90%). As sequence length increases, the basin of near-perfect reconstruction shrinks, making optimization harder. The first two PCA components explain 65.7% of the trajectory variance.

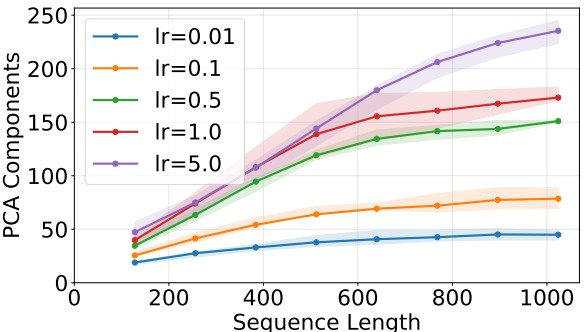

*Figure 4.* Number of principal components required to explain 99% of progressive trajectory variance vs. reconstructed prefix length for Llama-3.1-8B, shown for multiple learning rates.

to an affine rank-$k$ subspace. Empirically, this can improve stability and alter the effective capacity:

- It often reduces sensitivity to learning rate by concentrating optimization into a smaller coordinate system.

- It can reduce the intrinsic dimensionality of trajectories (PCA 99%) while consistently increasing information gain.

Table 3 compares baseline progressive cramming to a representative low-dimensional projection setting ("dim") for each family. We can see stable metrics growth across all models families.

**Is the low-rank subspace shared or sample-specific?**
The projection parameterization lets us probe whether valid compression embeddings inhabit a corpus-wide low-rank subspace or a sample-specific one. Training a single shared

*Table 3.* Progressive cramming variants across model families: baseline progressive cramming vs. low-dimensional projection only ("dim").

| Model | Compressed Tokens | Information Gain | Trajectory Length | PCA 99% |
|---|---|---|---|---|
| Llama-3.1-8B lr=0.1 | $1437.6 \pm 380.1$ | $4391 \pm 1408$ | $6174 \pm 1263$ | $82.78 \pm 9.07$ |
| Llama-3.1-8B dim=256 lr=0.1 | $1697.2 \pm 304.2$ | $4814 \pm 1731$ | $243424 \pm 57429$ | $58.58 \pm 6.4$ |
| pythia-1.4b lr=0.5 | $430.4 \pm 64.9$ | $1639 \pm 188$ | $6496 \pm 610$ | $50.06 \pm 3.31$ |
| pythia-1.4b dim=256 lr=0.5 | $500.2 \pm 74.8$ | $1874 \pm 283$ | $598025 \pm 725213$ | $67.58 \pm 19.82$ |
| SmolLM2-1.7B lr=0.1 | $335.1 \pm 61.3$ | $1208 \pm 162$ | $1051 \pm 147$ | $33.22 \pm 2.82$ |
| SmolLM2-1.7B dim=256 lr=0.1 | $957.4 \pm 142.5$ | $3271 \pm 309$ | $132821 \pm 44546$ | $30.94 \pm 4.93$ |
| gemma-3-4b-pt lr=0.1 | $213.6 \pm 109.2$ | $783 \pm 370$ | $910 \pm 261$ | $30.8 \pm 9.45$ |
| gemma-3-4b-pt dim=32 lr=0.1 | $697.4 \pm 165.3$ | $2141 \pm 786$ | $27948 \pm 15706$ | $22.5 \pm 5.14$ |

projection $(\mathbf{W}, \mathbf{b})$ jointly across 50 PG19 samples on SmolLM2-1.7B (per-sample $\mathbf{z}$, $k = 256$) reaches $742 \pm 177$ compressed tokens and $2566 \pm 511$ bits of information gain—only modestly below the per-sample projection ($957 \pm 142$ tokens, $3271 \pm 309$ bits), so a single basis can serve an entire corpus. However, freezing this shared basis and transferring it to 50 unseen PG19 samples—optimizing only $\mathbf{z}$—collapses capacity to $8.7 \pm 6.8$ tokens and $36 \pm 29$ bits. The shared basis thus succeeds only because $(\mathbf{W}, \mathbf{b})$ co-adapt with the specific coefficients seen during training; the low-rank subspace of valid solutions is sample-specific rather than a universal property of the model, consistent with the wide, near-orthogonal solution set discussed above.

### 5.3. PCA Reconstruction

We evaluate reconstruction accuracy when progressive embeddings are reconstructed from PCA components. Figure 5 reports accuracy as a function of the number of components. Although Table 3 suggests that for Llama-3.1-8B roughly 83 PCA components explain 99% of trajectory variance, achieving near-perfect teacher-forced reconstruction requires substantially more components. We also observe the same failure mode as in the full cramming setup (Section 3.2): errors cluster at the start of the sequence (often at the first decoded token), which then propagates and breaks subsequent autoregressive decoding.

## 6. Downstream Evaluation

We test whether cramming preserves the *useful* information in a prefix by evaluating standard multiple-choice benchmarks (HellaSwag and ARC) under likelihood-based scoring. These benchmarks use relatively short contexts, so they isolate capability failures that occur even when the task does not require long-range context retention. If the compression embedding faithfully encodes the prefix in a way the model can consume, accuracy should remain close to the uncompressed baseline; large drops indicate that conditioning on the compression embedding disrupts the model's ability to leverage the prefix for downstream reasoning.

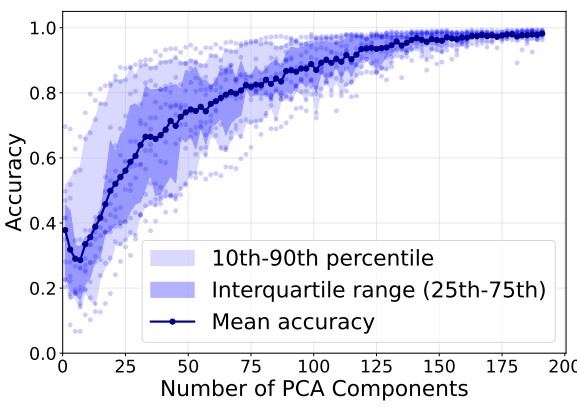

*Figure 5.* Teacher-forced reconstruction accuracy for PCA-reconstructed compression embeddings vs. number of PCA components (Llama-3.1-8B).

**Setup.** We evaluate on HellaSwag and ARC under two conditions:

1. Original prefix (baseline)

2. Compression embedding + original prefix

For the second condition, we optimize a separate compression embedding per benchmark instance, targeting perfect teacher-forced reconstruction of the corresponding prefix within a fixed optimization budget. We report both cramming variants: the standard full cramming setup (a single embedding optimized for the entire prefix) and the progressive stage-wise procedure from Section 4.1. Because the optimization does not always reach perfect reconstruction, we additionally report the fraction of benchmark instances that fully converged (per-sample token-argmax match rate of 1.0) and compute downstream accuracy over the converged subset only. This isolates the effect of conditioning on a faithfully reconstructed embedding from residual reconstruction errors. Each benchmark instance provides four candidate continuations; we select the continuation with

the lowest normalized negative log-likelihood (per suffix token). Under this 4-way multiple-choice protocol, random guessing yields 25% accuracy.

**Results.** Table 4 shows that prepending a crammed compression embedding consistently reduces downstream accuracy across model families, a moderate but reliable drop that nonetheless stays above chance on these multiple-choice benchmarks. Compression embeddings from progressive and full cramming perform comparably here. Together with the attention analysis, these results suggest that high reconstruction performance can coincide with brittle steering rather than a transferable semantic representation. This highlights a key limitation of using reconstruction accuracy alone: perfect reconstruction is insufficient, and future work must explicitly evaluate capability preservation in any downstream semantic evaluation and broader capability benchmarks overall. The drop reflects the compression embedding itself rather than residual reconstruction errors: because HellaSwag and ARC score fixed context–continuation pairs without regenerating the prefix, restricting the evaluation to perfectly reconstructed instances leaves accuracy essentially unchanged. This degradation becomes a complete collapse under a *generative* benchmark: on 5-shot MMLU (Appendix D.3, Table 23), compressing the full prefix drives accuracy to near zero with no valid parseable answers, while a random control embedding leaves accuracy nearly intact – confirming that the degradation is specifically caused by the optimized compression embedding rather than by conditioning on an out-of-distribution token.

# 7. Causal Analysis: Attention Mass vs. Causal Importance

The compression embedding becomes an attention sink, concentrating 20–60% of attention mass in specific intermediate layers (Appendix E.1). That analysis is correlational, however: it shows *where* the compression embedding accumulates attention, not whether that attention is *causally* responsible for the downstream degradation in Section 6. To establish causality, we intervene directly on the compression embedding using *attention knockout*: at selected layers we mask the pre-softmax attention logits that target the compression embedding (position 0), completely removing its influence on the residual stream at those layers while leaving all other computation intact. We use three protocols on Llama-3.1-8B ($L = 32$ layers):

1. **Per-layer knockout**: mask the compression embedding at a single layer $l$.

2. **Forward cumulative knockout**: mask layers 0 through $k$ (early-to-late).

3. **Reverse cumulative knockout**: mask layers $k$ through

$L - 1$ (late-to-early).

Each condition is evaluated on teacher-forced reconstruction accuracy and on downstream capability (HellaSwag), so we can ask whether a given layer's interaction with the compression embedding supports faithful reconstruction, downstream disruption, or both.

**Attention mass and causal importance are dissociated.** Figure 6 shows per-layer knockout. Knocking out the compression embedding at the earliest layers sharply reduces teacher-forced reconstruction accuracy: these are the layers where the embedding injects the information needed to reconstruct the prefix. The same early-layer interventions do not harm downstream accuracy; if anything they nudge it back toward the uncompressed baseline. By contrast, the layers that carry the most attention mass lie late in the network, yet knocking them out individually changes neither reconstruction nor downstream accuracy appreciably. The layers with the highest attention mass are therefore not the layers with the highest causal importance for downstream behavior.

**A forward/reverse asymmetry localizes the cause.** The cumulative protocols (Figure 7) make the locus of causal importance explicit. Forward cumulative knockout, which masks the earliest layers first, restores downstream accuracy to the uncompressed baseline after only the first several layers are masked – while progressively destroying reconstruction. Reverse cumulative knockout, which masks the latest layers first, leaves downstream accuracy near the crammed level across most of the network and recovers only once masking extends into the early layers. This asymmetry is the core causal claim: downstream collapse is driven by the compression embedding's interactions in the early layers, not by the late-layer attention concentration. The high attention mass observed in late layers is thus a *symptom* of cramming rather than the *cause* of capability loss; attention mass and causal importance are dissociated. This refines the picture: the compression embedding steers the model through early-layer computations that are essential for reconstruction but disruptive to the model's normal use of the prefix.

**The causal picture holds across model families.** We replicate all three knockout protocols on Pythia-1.4B and SmolLM2-1.7B and obtain consistent results: early-layer interactions causally drive the downstream collapse, the high-attention-mass late layers are not the causally important ones, and the forward/reverse cumulative asymmetry reappears in both models.

*Table 4.* Downstream multiple-choice evaluation on HellaSwag and ARC-Easy (4-way; chance = 25%). "Base" uses the original prefix and is evaluated on all samples. For each cramming variant (Progressive, Full), "Acc" reports accuracy over the converged subset only and "Conv%" reports the fraction of samples that fully converged. A sample is fully converged when its per-token argmax match rate equals 1.0 (token-perfect reconstruction). Empty cells ("–") indicate that the corresponding run does not exist or has zero converged samples (denominator 0).

| | HellaSwag | | | | | ARC-E | | | | |
| | | Progressive | | Full | | | Progressive | | Full | |
| Model | Base | Acc | Conv% | Acc | Conv% | Base | Acc | Conv% | Acc | Conv% |
| --- | --- | --- | --- | --- | --- | --- | --- | --- | --- | --- |
| pythia-1.4b | 44.00% | 37.63% | 97.07% | 37.83% | 97.07% | 53.10% | 49.11% | 98.63% | 50.50% | 98.63% |
| SLM2-1.7B | 53.56% | 37.01% | 95.51% | 38.71% | 96.88% | 67.26% | 56.00% | 97.66% | 53.29% | 97.85% |
| Llama-3.1-8B | 61.91% | 40.80% | 97.66% | 40.00% | 97.66% | 63.54% | 43.31% | 97.85% | 47.72% | 98.63% |
| gemma-3-4b | 57.07% | – | 0.00% | 46.67% | 5.86% | 64.58% | 59.38% | 6.25% | 54.17% | 4.69% |

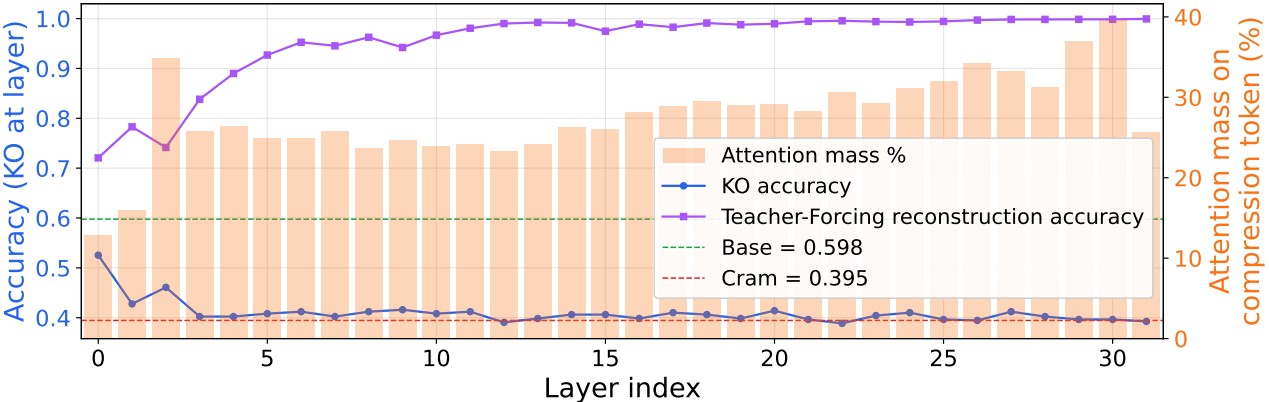

*Figure 6.* Per-layer attention knockout on Llama-3.1-8B. Masking the compression embedding at a single layer, we report downstream HellaSwag accuracy (blue) and teacher-forced reconstruction accuracy (purple); orange bars show the compression embedding's attention mass per layer. Dashed lines mark the uncompressed baseline (Base) and the crammed prefix without intervention (Cram). Early-layer knockout sharply degrades reconstruction yet leaves downstream accuracy at or above the crammed baseline, whereas the late layers that carry the most attention mass have negligible causal effect on either metric.

## 8. Compression Capacity Scales with Depth and Model Size

How much a single embedding can cram is not fixed: it is bounded by the capacity of the frozen model that must reconstruct from it. We probe that bound directly by truncating pretrained models to their first $N$ decoder layers, repairing the cut with a short causal-LM finetune on fineweb-edu, and running the standard progressive-cramming evaluation (PG19, 50 samples, $\mathrm{lr} = 0.1$) on each truncated checkpoint. Sweeping $N \in \{1, 2, 4, 8\}$ and the full model across five families that span 1.7–8B parameters and 24–36 layers (Table 5) separates two axes of capacity: network *depth* (within a family) and model *size* (across families at a matched truncated depth). The Qwen3 (4B, 8B) and SmolLM3-3B families appear only in this capacity experiment; SmolLM2-1.7B and Llama-3.1-8B are shared with our main evaluation (Section 5).

Both axes drive compression monotonically, and they compound. **Depth:** within every family the number of perfectly crammed tokens rises with retained depth—roughly dou-

bling per layer-doubling (e.g. Qwen3-8B: $119 \rightarrow 180 \rightarrow 304 \rightarrow 625$ at first-1/2/4/8). **Size:** at any matched truncated depth the larger model crams strictly more (Qwen3-8B > Qwen3-4B > SmolLM3-3B at every $N$). The two effects reinforce one another: with enough width a shallow truncation already recovers most of the full-model capacity—the 8-of-36-layer Qwen3-8B and Qwen3-4B reach $\approx$81–82% of their untruncated references, and the 8-layer Qwen3-8B exceeds the *full* 36-layer Qwen3-4B. Capacity is thus jointly governed by depth and width—intuitively, depth unpacks the crammed code while width stores it—so a deficit in one can be partly offset by a surplus in the other.

Because every truncated checkpoint is repaired with the same short finetune before we measure it, this monotone trend reflects the retained network's capacity rather than the transient damage of the cut itself. The practical reading is that compressibility is an architectural property the reconstructor can be designed for: trading depth against width, or adding either, raises the ceiling on how much a single embedding can hold, so a model can be made more

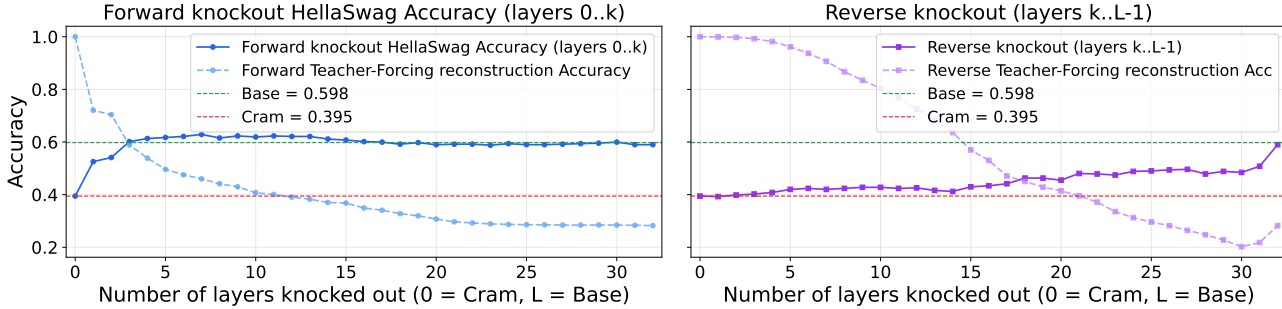

*Figure 7.* Cumulative attention knockout on Llama-3.1-8B. Left: forward knockout masks layers 0 through $k$; right: reverse knockout masks layers $k$ through $L-1$. Curves show downstream HellaSwag accuracy and teacher-forced reconstruction accuracy as a function of the number of knocked-out layers (0 = crammed input, $L$ = uncompressed baseline). Forward knockout restores downstream accuracy to the baseline after masking only the first few layers, whereas reverse knockout fails to recover until masking finally reaches the early layers – an asymmetry showing that early-layer interactions causally drive the downstream collapse.

*Table 5.* Compression capacity (mean perfectly crammed tokens over 50 PG19 samples) vs. model and retained depth. Columns give the number of first decoder layers kept (then finetuned); "Full" is the untruncated model; each model name carries its total decoder-layer count. "–" marks configurations we did not run. Compression rises with both retained depth (left to right) and model size (top to bottom).

| Model | Retained first-$N$ layers | | | | Full |
|---|---|---|---|---|---|
| | 1 | 2 | 4 | 8 | |
| SmolLM2-1.7B (24L) | 50 | 241 | 404 | 455 | 335 |
| SmolLM3-3B (36L) | 20 | 39 | 114 | 300 | – |
| Qwen3-4B (36L) | 79 | 147 | 192 | 421 | 512 |
| Qwen3-8B (36L) | 119 | 180 | 304 | 625 | 774 |
| Llama-3.1-8B (32L) | 97 | 184 | 400 | – | 1438 |

compressible without simply being made larger. The substitution is only partial, however: at the shallowest truncations even the widest model recovers far less than its full-depth reference, so width cannot fully stand in for the missing layers that unpack the code. Taken together, these axes suggest that the compression ceiling tracks total reconstructor capacity rather than any single dimension of it.

## 9. Conclusion

We introduced progressive cramming, which grows the reconstruction target token-by-token and enables controlled measurement of when embedding-based reconstruction succeeds or fails under a fixed optimization budget. Across model families, progressive trajectories are low-dimensional. This low dimensionality is a property of the warm-started optimization *path* rather than of an intrinsically small solution set: embeddings optimized independently for the same prefix end up far apart and nearly orthogonal. Causal attention-knockout interventions localize the downstream collapse to the compression embedding's interactions in the early layers, and show that the late layers carrying the most attention mass have little causal effect – attention mass is

a symptom rather than the cause of capability loss. Separately, we find that how much a single embedding can store is bounded by the frozen reconstructor itself: compression capacity rises monotonically with both retained model depth and width. Because the two axes partly substitute for one another, it is the reconstructor's architecture—not parameter count alone—that sets the compression ceiling, so compressibility can be improved by reshaping the model and not only by scaling it. Critically, downstream evaluation indicates that optimizing embeddings for reconstruction can coincide with large capability drops, even when the original prefix remains in context, so reconstruction accuracy alone is not a sufficient criterion for meaningful compression.

Our study leaves several questions open, each pointing to a concrete direction for future work. First, we do not yet know how to make a single embedding both perfectly reconstructable and semantically faithful; reconciling these two objectives remains the central open problem. Second, progressive cramming is costly: it is roughly twice as slow as full cramming and resists batching, since each sample reaches perfect reconstruction at its own rate. A fast compressor that produced semantically meaningful embeddings in a single forward pass, without per-sample backpropagation, would remove this bottleneck. Third, predicting the reconstruction boundary in advance would speed up convergence; information gain is the most reliable proxy we found, but it is still too coarse to pinpoint a model's capacity exactly. Finally, our low-dimensional-projection results suggest that better optimizers could compress longer sequences than we reach here.

## Impact Statement

This paper presents work whose goal is to advance the field of Machine Learning. There are many potential societal consequences of our work, none which we feel must be specifically highlighted here.

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

# A. Experimental Details

## A.1. Hyperparameters

We used 50 PG19 samples for most experiments in the main paper body from `LarryLovestein/pg19_1k` that were repoduced by the recipie from (Kuratov et al., 2025). While some results in appendix were reported only for 10 PG19 samlpes from that dataset. Across all settings, the base language model was frozen; only the compression-embedding parameters were optimized (either the compression embeddings, or the low-dimensional embedding together with its projection). We optimized with AdamW. For each model, we swept the learning rate over $\{0.01, 0.1, 0.5, 1.0\}$; the resulting metrics are summarized in Table 7. For the low-dimensional embedding and projection variant, we used a learning rate of 0.01. We ran up to 10,000 optimization steps per sample in all experiments; for progressive runs, we additionally capped the optimization of each newly added token at 1,000 steps. We used 100 warmup steps for all models.

## A.2. Compute

Each experiment ran on a single NVIDIA A100 80GB GPU, with up to 16 GPUs used in parallel across experiments. Training time depends on the sequence length and loss type, ranging from a few minutes (short sequences, small models) to approximately 24 hours (long sequences, large models). The total compute used for this research (including debugging and unsuccessful experiments) is approximately 2800 GPU-hours.

# B. Progressive Cramming: Variants and Regularizers

## B.1. Full cramming vs progressive.

In Table 6 we present results from Section 5 for full and progressive cramming comparison for different model sizes of Llama and Pythia families.

## B.2. Progressive across models scales

Table 8 shows that larger models tend to use more trajectory directions (higher PCA 99%).

## B.3. Tokens-Per-Stage Ablation

Progressive cramming grows the target prefix one token at a time: each time the current stage converges (its compression embedding perfectly reconstructs the prefix), the procedure appends a single token to the prefix and re-optimizes. Here we ablate this growth granularity on SmolLM2-1.7B by varying the *progressive step* $\Delta$—the number of target tokens appended to the prefix each time a stage converges,

*Table 6.* Full vs. progressive cramming on PG19 (mean ± std over samples). "Full" cramming (Kuratov et al., 2025) uses a fixed token budget and may stop below perfect Teacher Forcing (TF) reconstruction; progressive cramming fixes reconstruction at 100% and guarantees perfect reconstruction under autoregressive greedy generation.

| Type | Tokens | Accuracy (%) | |
| | | TF | Greedy |
|---|---|---|---|
| **Llama-3.2-1B** | | | |
| Full | 512 | 99.80 ± 0.00 | 1.51 ± 0.94 |
| Prog. | 402 ± 85 | 100.00 | 100.00 |
| **Llama-3.2-3B** | | | |
| Full | 1024 | 99.60 ± 0.70 | 1.02 ± 1.23 |
| Prog. | 902 ± 207 | 100.00 | 100.00 |
| **Llama-3.1-8B** | | | |
| Full | 1568 | 99.72 ± 0.32 | 1.29 ± 1.12 |
| Prog. | 1064 ± 394 | 100.00 | 100.00 |
| **Pythia160m** | | | |
| Full | 32 | 68.44 ± 17.54 | 26.39 ± 24.08 |
| Prog. | 11 ± 2 | 100.00 | 100.00 |
| **Pythia410m** | | | |
| Full | 128 | 89.06 ± 6.15 | 13.95 ± 23.93 |
| Prog. | 102 ± 41 | 100.00 | 100.00 |
| **Pythia1.4b** | | | |
| Full | 256 | 98.48 ± 3.06 | 39.23 ± 43.79 |
| Prog. | 544 ± 57 | 100.00 | 100.00 |

and which the compression embedding must then jointly reconstruct—over $\Delta \in \{1, 2, 4, 8, 16, 32, 64, 128\}$. The default $\Delta = 1$ (grow one token at a time) is the reference; for $\Delta > 1$ we also start the schedule at a prefix length of $\Delta$ so the prefix is always a multiple of $\Delta$. Every other factor is held fixed at the main run's configuration (cross-entropy reconstruction, single-layer activation alignment, no low-dimensional projection, the same PG19 sample set, learning rate 0.1, and optimization budget).

Because the only varying factor is how many new tokens each converged stage must absorb at once, this ablation isolates the role of the fine-grained curriculum: a larger $\Delta$ asks the compression embedding to make a bigger jump per stage rather than a sequence of single-token extensions. Table 9 reports, for each $\Delta$, the achieved compressed-token count, information gain, trajectory length, the number of principal components explaining 99% of trajectory variance (PCA 99%), and the cumulative number of optimization steps spent to reach the last successfully compressed token. The "Compressed Tokens" column is the largest *converged* prefix length $n$, i.e. the longest prefix the compression embedding reconstructs perfectly; progressive cramming halts at the first stage that fails to converge, so we exclude that final non-converged stage—counting it would inflate $n$ by up to $\Delta$ tokens and spuriously favour large steps.

*Table 7.* Results of the learning-rate sweep across models and settings. "Compressed Tokens" is the achieved prefix length $n$, "Trajectory Length" is $L_{\text{traj}}$, and "PCA 99%" is the number of principal components explaining 99% of trajectory variance.

| Model | Compressed Tokens | Information Gain | Trajectory Length | PCA 99% |
|---|---|---|---|---|
| Llama-3.1-8B | $1298.4 \pm 537.5$ | $3760 \pm 1418$ | $738 \pm 265$ | $40.8 \pm 13.16$ |
| Llama-3.1-8B lr=0.1 | $1063.5 \pm 394.4$ | $3028 \pm 1321$ | $4861 \pm 1033$ | $74.4 \pm 7.94$ |
| Llama-3.1-8B lr=0.5 | $934.4 \pm 123.2$ | $2758 \pm 486$ | $15424 \pm 1437$ | $138 \pm 11.78$ |
| Llama-3.1-8B lr=1.0 | $1286.4 \pm 524.8$ | $3381 \pm 1521$ | $29061 \pm 4605$ | $186.6 \pm 13.59$ |
| pythia-1.4b | $188.2 \pm 27.8$ | $581 \pm 101$ | $83 \pm 14$ | $15.7 \pm 1.85$ |
| pythia-1.4b lr=0.1 | $274.3 \pm 56.2$ | $846 \pm 106$ | $956 \pm 123$ | $23.2 \pm 1.99$ |
| pythia-1.4b lr=0.5 | $543.8 \pm 56.8$ | $1694 \pm 125$ | $6730 \pm 390$ | $49 \pm 3.32$ |
| pythia-1.4b lr=1.0 | $558.2 \pm 74.1$ | $1737 \pm 158$ | $11381 \pm 993$ | $63.3 \pm 6.21$ |
| SmolLM2-1.7B lr=0.1 | $370.2 \pm 113.1$ | $1119 \pm 350$ | $1027 \pm 167$ | $29.2 \pm 2.93$ |
| SmolLM2-1.7B lr=0.5 | $540.5 \pm 139.4$ | $1450 \pm 350$ | $5225 \pm 827$ | $66.8 \pm 9.14$ |
| SmolLM2-1.7B lr=1.0 | $502.5 \pm 180.6$ | $1428 \pm 473$ | $9442 \pm 1810$ | $83 \pm 16.34$ |
| gemma-3-4b-pt lr=0.1 | $286.5 \pm 164.6$ | $949 \pm 466$ | $1044 \pm 328$ | $31.8 \pm 9.21$ |
| gemma-3-4b-pt lr=0.5 | $498.6 \pm 86.5$ | $1437 \pm 308$ | $6055 \pm 925$ | $71.6 \pm 11.34$ |
| gemma-3-4b-pt lr=1.0 | $558 \pm 86.1$ | $1258 \pm 511$ | $12173 \pm 1004$ | $101.6 \pm 10.66$ |

*Table 8.* Progressive cramming trajectory statistics on PG19. "Compressed Tokens" is the achieved prefix length $n$, "Trajectory Length" is $L_{\text{traj}}$, and "PCA 99%" is the number of principal components explaining 99% of trajectory variance.

| Model | Compressed Tokens | Information Gain | Trajectory Length | PCA 99% |
|---|---|---|---|---|
| Llama-3.2-1B lr=0.1 | $402.2 \pm 84.8$ | $1500 \pm 282$ | $1736 \pm 206$ | $44.1 \pm 5.73$ |
| Llama-3.2-3B lr=0.1 | $902.2 \pm 206.8$ | $3074 \pm 527$ | $4232 \pm 913$ | $61.5 \pm 4.54$ |
| Llama-3.1-8B lr=0.1 | $1063.5 \pm 394.4$ | $3028 \pm 1321$ | $4861 \pm 1033$ | $74.4 \pm 7.94$ |
| pythia-160m lr=0.5 | $10.7 \pm 2$ | $19 \pm 26$ | $652 \pm 196$ | $5.1 \pm 1.64$ |
| pythia-410m lr=0.5 | $102.1 \pm 40.8$ | $323 \pm 105$ | $3613 \pm 841$ | $37.5 \pm 11.16$ |
| pythia-1.4b lr=0.5 | $543.8 \pm 56.8$ | $1694 \pm 125$ | $6730 \pm 390$ | $49 \pm 3.32$ |
| SmolLM2-135M lr=0.1 | $38.5 \pm 14.1$ | $168 \pm 66$ | $178 \pm 40$ | $12.2 \pm 2.96$ |
| SmolLM2-360M lr=0.1 | $61 \pm 24.6$ | $266 \pm 122$ | $234 \pm 111$ | $13.4 \pm 3.95$ |
| SmolLM2-1.7B lr=0.1 | $370.2 \pm 113.1$ | $1119 \pm 350$ | $1027 \pm 167$ | $29.2 \pm 2.93$ |
| gemma-3-270m lr=0.1 | $71.2 \pm 14.2$ | $392 \pm 71$ | $259 \pm 32$ | $18.5 \pm 2.54$ |
| gemma-3-1b-pt lr=0.1 | $74.8 \pm 34.3$ | $338 \pm 104$ | $386 \pm 89$ | $18.9 \pm 6.59$ |
| gemma-3-4b-pt lr=0.1 | $286.5 \pm 164.6$ | $949 \pm 466$ | $1044 \pm 328$ | $31.8 \pm 9.21$ |

The achieved length does not grow monotonically with $\Delta$: it stays close to the $\Delta = 1$ baseline, with at most a mild peak at intermediate steps before returning to the baseline level at $\Delta = 128$. What does change markedly is the cost and shape of the trajectory: as $\Delta$ grows the last converged token is reached in substantially fewer optimization steps, and both the trajectory length $L_{\text{traj}}$ and PCA 99% fall sharply (the run takes fewer, larger stages). In other words, coarser per-stage growth is roughly as effective at compression while reaching the limit more cheaply and along a shorter, lower-dimensional path. Concretely, the mean number of perfectly reconstructed (compressed) tokens is 334.1 (1 token/stage), 340.9 (2 tokens/stage), 337.6 (4 tokens/stage), 351.2 (8 tokens/stage), 362.6 (16 tokens/stage), 367.4 (32 tokens/stage), 363.5 (64 tokens/stage), and 335.4 (128 tokens/stage) for the fixed-step arms, and 262.3, 240.8, and 272.7 for the geometric bisect, bisect-with-warm-restore, and linear-with-warm-restore back-offs respectively.

A complementary set of arms replaces the fixed step with an *adaptive* one. Rather than appending a constant number of tokens per converged stage, the *geometric growth* arms double the prefix length each time a stage converges—a geometrically growing number of added tokens, warm-started from the previous converged embedding. The first stage that fails to converge brackets the horizon between the largest converged length lo and the smallest failed length hi; a back-off phase then pins the exact largest converged prefix within that bracket. We compare three back-off strategies (last three rows of Table 9): **(i) bisect** halves the $(\text{lo}, \text{hi})$ gap, with each probe inheriting the embedding and optimizer state of the preceding (failed, longer) probe; **(ii) bisect + warm-restore** is identical but restores the last *converged* embedding, Adam moments, and learning-rate position before every probe; **(iii) linear + warm-restore** restores the converged anchor once and then grows the prefix one token per stage until a stage fails, mirroring the $\Delta = 1$ schedule but only over the bracket. Because the reported length only

ever advances on a converged stage, all three preserve progressive cramming's guarantee that the returned prefix is fully reconstructed; every stage uses the same per-token and cumulative per-sample optimization budgets as the fixed-step arms, so "Steps to Converge" is directly comparable.

All three adaptive arms reach the horizon in far fewer cumulative steps than the $\Delta = 1$ baseline (3.6–6.0k versus 8.8k), but all three *undershoot* its compressed-token count: 262 (bisect), 241 (bisect + warm-restore), and 273 (linear + warm-restore) versus 334 for $\Delta = 1$—an 18–28% shortfall. The fine-grained single-token curriculum is therefore not redundant: the patient one-token-at-a-time schedule locates a longer horizon than any logarithmic search, suggesting that the large doubling jumps land in a harder optimization regime that the bounded per-stage budget cannot always recover. Warm-restoring the converged anchor before each bisection probe does not help—it is slightly *worse* (241 versus 262), suggesting that the converged short-prefix optimizer state is a poor initialization for a much longer probe—whereas the gentle linear +1 back-off recovers the most (273), at the cost of more steps and a longer, higher-dimensional trajectory whose $L_{\text{traj}}$ and PCA 99% approach the $\Delta = 1$ values. In short, geometric search trades horizon quality for a large step saving, and the back-off strategy mediates that trade-off.

## B.4. Fixed-Prefix Progressive Cramming Ablation

Our main progressive-cramming setup asks a single compression embedding to reconstruct a sequence from scratch, with no other context. Here we ask how the task changes when the model is additionally given a fixed, *uncompressed* prefix that it can attend to but never has to compress. Concretely, we tokenize each document as $[\,p_1 \ldots p_P \mid c_1 \ldots c_L\,]$ and feed the model $[\,[\text{mem}]\ p_1 \ldots p_P\ c_1 \ldots c_L\,]$: the $P$ prefix tokens $p_{1:P}$ are real token embeddings the model reads as ordinary context, the learnable $[\text{mem}]$ token sits ahead of them, and the loss, convergence check, and information gain are computed only over the continuation $c_{1:L}$ that follows the prefix. The prefix is never folded into $[\text{mem}]$. Progressive cramming then grows the continuation $c_{1:L}$ token-by-token exactly as in the prefix-free setting, so "Compressed Tokens" counts only the crammed continuation, excluding the visible prefix. We sweep the prefix length $P \in \{128, 256, 512, 1024\}$ on SmolLM2-1.7B, with the no-prefix run ($P = 0$) as the reference.

All runs use the same PG19 sample set, learning rate (0.1), and optimization budget as our main SmolLM2-1.7B run, so the only varying factor is the prefix length $P$. Table 10 reports, for each $P$, the achieved compressed (continuation) token count, the information gain of the compression embedding measured on top of the prefix context, trajectory length, the number of principal components explaining 99%

of trajectory variance (PCA 99%), and the average base-LM surprisal over the prefix tokens themselves (bits per token). Concretely, the mean number of perfectly reconstructed (compressed) tokens is 361.5 (p=128), 352.8 (p=256), 363.5 (p=512), 360.2 (p=1024).

The prefix becomes more predictable as it lengthens: the base model's average surprisal over the prefix tokens falls monotonically from 4.09 bits/token at $P=128$ to 3.84, 3.73, and 3.62 bits/token at $P=256, 512, 1024$, consistent with a longer span of coherent context making each prefix token easier to anticipate. The cramming task itself, however, is left essentially unchanged. The number of continuation tokens the $[\text{mem}]$ token can perfectly cram is flat across prefix lengths ($\approx 360$ for every $P$, versus 335 with no prefix), and so are the information gain ($\approx 1150$–$1255$ bits), the trajectory length ($\approx 970$–$1050$), and the dimensionality of the optimization path (PCA-99% $\approx 33$ for all rows, including the reference). In other words, giving the model a large block of visible, uncompressed context neither eases nor reshapes the problem of packing the *continuation* into a single embedding: the compression embedding's capacity and the geometry of its optimization are governed by the span it must itself encode, not by how much surrounding text the model can already attend to.

## B.5. Activation Alignment

To stabilize optimization, we regularize hidden states toward those of the uncompressed sequence. Let $\mathbf{h}_l^{\text{cram}}(i)$ denote the hidden state at layer $l$ and token position $i$ when the model is conditioned on the compression embedding, and let $\mathbf{h}_l^{\text{clean}}(i)$ be the corresponding state for the same token position without compression. We align activations using cosine distance:

$$\mathcal{L}_{\text{align}} = \frac{1}{n} \sum_{l=1}^{K} \sum_{i=1}^{n} \left( 1 - \frac{\mathbf{h}_l^{\text{cram}}(i) \cdot \mathbf{h}_l^{\text{clean}}(i)}{\|\mathbf{h}_l^{\text{cram}}(i)\| \, \|\mathbf{h}_l^{\text{clean}}(i)\|} \right) \quad (6)$$

where $K$ is the number of layers used for alignment. We align the first $K$ transformer layers (closest to the embedding layer).

The full objective combines reconstruction and alignment:

$$\mathcal{L} = \mathcal{L}_{\text{cram}} + \alpha \mathcal{L}_{\text{align}} \quad (7)$$

where $\alpha$ controls alignment strength.

Table 11 studies activation alignment and its interaction with low-dimensional projection. Alignment acts as a regularizer: it can significantly reduce trajectory length and PCA 99% (indicating smoother, more constrained optimization), but it may trade off against raw cramming capacity depending on the model family. In particular, for Llama-3.1-8B, alignment increases information gain and token count relative to

*Table 9.* Tokens-per-stage ablation on SmolLM2-1.7B. We vary the progressive step $\Delta$—the number of target tokens appended to the prefix each time a stage converges—over $\{1, 2, 4, 8, 16, 32, 64, 128\}$; $\Delta = 1$ (grow one token at a time) is the reference. All runs share the same progressive eval configuration (PG19, sequence length 4096, learning rate 0.1, 50 samples). "Compressed Tokens" is the largest *converged* prefix length $n$ (the final non-converged stage is excluded, so it is not inflated by $\Delta$), "Trajectory Length" is $L_{\text{traj}}$, "PCA 99%" is the number of principal components explaining 99% of trajectory variance, and "Steps to Converge" is the cumulative optimization steps spent to reach that last converged token. The final three rows (*geometric growth*) replace the fixed step with a doubling schedule and one of three back-off strategies—bisection, bisection with warm-restore of the last converged state, or a linear $+1$ walk with warm-restore—each locating the horizon in far fewer stages while still guaranteeing the returned prefix is fully reconstructed, though at a compressed-token count below the $\Delta = 1$ reference.

| Model | Compressed Tokens | Information Gain | Trajectory Length | PCA 99% | Steps to Converge |
|---|---|---|---|---|---|
| 1 token/stage | $334.1 \pm 61.3$ | $1208 \pm 162$ | $1051 \pm 147$ | $33.22 \pm 2.82$ | $8825 \pm 2605$ |
| 2 tokens/stage | $340.9 \pm 51.5$ | $1250 \pm 179$ | $772 \pm 94$ | $30.1 \pm 2.6$ | $9151 \pm 1298$ |
| 4 tokens/stage | $337.6 \pm 62.6$ | $1238 \pm 158$ | $555 \pm 67$ | $26.72 \pm 2.17$ | $9333 \pm 456$ |
| 8 tokens/stage | $351.2 \pm 57$ | $1287 \pm 181$ | $423 \pm 48$ | $21.98 \pm 1.95$ | $8871 \pm 663$ |
| 16 tokens/stage | $362.6 \pm 62$ | $1344 \pm 141$ | $311 \pm 36$ | $15.58 \pm 1.67$ | $8696 \pm 778$ |
| 32 tokens/stage | $367.4 \pm 55.9$ | $1400 \pm 130$ | $226 \pm 25$ | $9.6 \pm 1.26$ | $7947 \pm 1130$ |
| 64 tokens/stage | $363.5 \pm 69.5$ | $1453 \pm 163$ | $149 \pm 19$ | $5.18 \pm 0.99$ | $6971 \pm 1459$ |
| 128 tokens/stage | $335.4 \pm 91.9$ | $1468 \pm 223$ | $83 \pm 16$ | $2.42 \pm 0.6$ | $5627 \pm 2673$ |
| geometric (bisect) | $262.3 \pm 52.4$ | $958 \pm 128$ | $209 \pm 30$ | $7.64 \pm 0.93$ | $3613 \pm 1177$ |
| geometric (bisect+restore) | $240.8 \pm 53$ | $882 \pm 132$ | $281 \pm 37$ | $9.5 \pm 0.96$ | $4773 \pm 1326$ |
| geometric (linear+restore) | $272.7 \pm 55.5$ | $1002 \pm 151$ | $611 \pm 66$ | $26.56 \pm 3.41$ | $6045 \pm 1834$ |

*Table 10.* Fixed-prefix progressive cramming ablation on SmolLM2-1.7B. The model attends to a fixed, uncompressed prefix of $P$ real tokens and crams only the continuation that follows; the $P = 0$ row is the no-prefix reference. All rows use the identical progressive-cramming eval (PG19, lr $= 0.1$). "Compressed Tokens" is the achieved continuation length $n$ (excluding the prefix), "Trajectory Length" is $L_{\text{traj}}$, "PCA 99%" is the number of principal components explaining 99% of trajectory variance, and "Avg Prefix Surprisal" is the base-LM next-token cross-entropy over the prefix tokens, in bits per token.

| Model | Compressed Tokens | Information Gain | Trajectory Length | PCA 99% | Avg Prefix Surprisal (bits/tok) |
|---|---|---|---|---|---|
| No prefix | $335.1 \pm 61.3$ | $1208 \pm 162$ | $1051 \pm 147$ | $33.22 \pm 2.82$ | – |
| P=128 | $361.5 \pm 78.4$ | $1255 \pm 240$ | $977 \pm 159$ | $33.38 \pm 3.68$ | $4.09 \pm 0.63$ |
| P=256 | $352.8 \pm 87.9$ | $1205 \pm 234$ | $967 \pm 176$ | $33.98 \pm 3.8$ | $3.84 \pm 0.51$ |
| P=512 | $363.5 \pm 96.8$ | $1206 \pm 278$ | $1010 \pm 220$ | $33.08 \pm 4.84$ | $3.73 \pm 0.4$ |
| P=1024 | $360.2 \pm 98.3$ | $1154 \pm 247$ | $1022 \pm 268$ | $33.78 \pm 4.94$ | $3.62 \pm 0.37$ |

the baseline, whereas for SmolLM2-1.7B and Pythia-1.4B alignment alone reduces capacity but can be combined with projection to recover part of the loss. Across settings, we observe that alignment qualitatively changes attention behavior, often pushing attention-concentration effects toward deeper layers.

### B.6. Full activation alignment experiments

Table 12 reports the full sweep over activation alignment strength $\alpha$, number of aligned layers $L$, and projection dimension, together with the resulting progressive cramming metrics.

### B.7. Full Low dimensional experiments

Table 13 ablates projection dimension across model families. Three takeaways stand out:

- For Llama-3.1-8B and SmolLM2-1.7B, projection can substantially increase achievable token counts and information gain compared to the unconstrained baseline.

- Increasing projection dimension does not reliably increase PCA 99% (and can even decrease it), but it often increases the measured trajectory length, suggesting longer optimization paths within a similarly low-dimensional subspace.

- For Pythia-1.4B, small projection dimensions strongly reduce PCA 99% and trajectory length, but also reduce capacity; adding a larger projection partially recovers information gain.

### B.8. Prefix tuning

Prefix tuning introduces a separate trainable prefix embedding at each transformer layer, avoiding the per-layer forward-pass bottleneck of soft-prompt setups such as full or progressive cramming. The trade-off is parameter cost: it requires $L$ times more trainable embeddings, where $L$ is the number of hidden layers in the LLM.

To separate "cramming" from general soft-prompt steering, we evaluate prefix tuning baselines. Table 14 reports the same attention-mass metric for prefix tokens. Compared to

*Table 11.* Progressive cramming variants and key metrics across model families. For each family we report baseline progressive cramming, low-dimensional projection only ("dim"), activation alignment only ($\alpha$, $L$), and their combination.

| Model | Compressed Tokens | Information Gain | Trajectory Length | PCA 99% |
|---|---|---|---|---|
| Llama-3.1-8B lr=0.1 | $1437.6 \pm 380.1$ | $4391 \pm 1408$ | $6174 \pm 1263$ | $82.78 \pm 9.07$ |
| Llama-3.1-8B dim=256 lr=0.1 | $1697.2 \pm 304.2$ | $4814 \pm 1731$ | $243424 \pm 57429$ | $58.58 \pm 6.4$ |
| Llama-3.1-8B lr=0.1 $\alpha = 1.0$ $L = 4$ | $1795.1 \pm 225.7$ | $5605 \pm 596$ | $7525 \pm 774$ | $81.26 \pm 3.76$ |
| Llama-3.1-8B dim=256 lr=0.1 $\alpha = 1.0$ $L = 8$ | $1732.7 \pm 253.2$ | $4009 \pm 2414$ | $226998 \pm 45513$ | $62.38 \pm 5.08$ |
| pythia-1.4b lr=0.5 | $430.4 \pm 64.9$ | $1639 \pm 188$ | $6496 \pm 610$ | $50.06 \pm 3.31$ |
| pythia-1.4b lr=0.5 $\alpha = 1.0$ $L = 8$ | $397.7 \pm 57$ | $1513 \pm 163$ | $6187 \pm 513$ | $46.44 \pm 2.74$ |
| pythia-1.4b dim=256 lr=0.5 | $500.2 \pm 74.8$ | $1874 \pm 283$ | $598025 \pm 725213$ | $67.58 \pm 19.82$ |
| pythia-1.4b dim=256 lr=0.5 $\alpha = 1.0$ $L = 8$ | $405.3 \pm 82.5$ | $1521 \pm 276$ | $1269523 \pm 618392$ | $58.92 \pm 15.36$ |
| SmolLM2-1.7B lr=0.1 | $335.1 \pm 61.3$ | $1208 \pm 162$ | $1051 \pm 147$ | $33.22 \pm 2.82$ |
| SmolLM2-1.7B lr=0.1 $\alpha = 1.0$ $L = 8$ | $242.6 \pm 63.2$ | $880 \pm 186$ | $894 \pm 121$ | $26.46 \pm 2.64$ |
| SmolLM2-1.7B dim=256 lr=0.1 | $957.4 \pm 142.5$ | $3271 \pm 309$ | $132821 \pm 44546$ | $30.94 \pm 4.93$ |
| SmolLM2-1.7B dim=256 lr=0.1 $\alpha = 1.0$ $L = 8$ | $918.6 \pm 199.1$ | $3135 \pm 628$ | $142874 \pm 54379$ | $33.28 \pm 6.69$ |
| gemma-3-4b-pt lr=0.1 | $213.6 \pm 109.2$ | $783 \pm 370$ | $910 \pm 261$ | $30.8 \pm 9.45$ |
| gemma-3-4b-pt lr=0.1 $\alpha = 1.0$ $L = 8$ | $202 \pm 119.4$ | $763 \pm 425$ | $869 \pm 275$ | $26.86 \pm 7.7$ |
| gemma-3-4b-pt dim=32 lr=0.1 | $697.4 \pm 165.3$ | $2141 \pm 786$ | $27948 \pm 15706$ | $22.5 \pm 5.14$ |
| gemma-3-4b-pt dim=32 lr=0.1 $\alpha = 1.0$ $L = 8$ | $680.7 \pm 221$ | $2091 \pm 730$ | $35384 \pm 29987$ | $21.78 \pm 4.94$ |

cramming, prefix tuning typically attracts less attention mass than BOS and shows weaker or more variable correlation patterns, indicating that this attention concentration is not an inevitable consequence of prepending learned embeddings but depends on the optimization objective.

We also evaluated compression limits under prefix tuning. Table 15 shows that, for Llama 3 models at 3B and 8B, we cannot reach the cramming limits; however, in the prefix-tuning setup these models can still compress up to 8k tokens with perfect reconstruction accuracy, while 16k tokens results in out-of-memory (OOM) failures. For the 1B Llama 3 model, the limit is about 4096 tokens, which is roughly $10\times$ higher than with progressive cramming. This gain comes at a higher parameter cost: prefix tuning requires $L$-times more trainable embeddings (one per layer), i.e., $16\times$ more parameters for a 16-layer model. For Pythia, the prefix-tuning setup increases the compression limit by about $20\times$ for Pythia-410M and by about $8\times$ for Pythia-1.4B relative to progressive cramming.

## C. Optimization Geometry and Trajectory Structure

### C.1. Dataset Modifications

To probe sensitivity to surface form, we create controlled variants of PG19 samples and compare their progressive trajectories (Figure 8). Each variant shares an identical 64-token prefix and modifies the suffix in one of three ways: random word permutation, greedy decoding from Llama-3.1-8B ("Sampled"), or lowercasing. Trajectories coincide during the shared prefix and diverge immediately after the edit point, indicating that the optimization path is highly

sensitive to token-level details. Notably, lowercasing (which should preserve high-level semantics) induces a markedly different trajectory, suggesting that progressive cramming is driven by tokenization- and distribution-level properties rather than semantic equivalence.

### C.2. Surprisal Predicts Per-Token Cramming Effort

Progressive cramming spends a variable number of optimizer steps on each token: some are absorbed in a handful of steps, others take hundreds. This section asks what governs that per-token cost, and finds that it is well predicted by a single quantity computed from the *frozen* base model—the next-token surprisal of the token being added. With the default single-token schedule ($\Delta = 1$), each stage grows the target prefix by exactly one token and re-optimizes the compression embedding (warm-started from the previous, already-converged stage) until it reconstructs the prefix. The optimizer steps the stage at prefix length $n$ spends to absorb that one new token is therefore the marginal cramming cost of token $n$, which we denote $\Delta\text{steps}(n)$. We compare it against the base model's per-token surprisal $s(n) = -\log_2 p(x_n \mid x_{<n})$ in bits, i.e. the per-token quantity whose running sum is the description length $\text{DL}(n) = \sum_{i \le n} s_i$; its total $\text{DL}(N)$ is exactly the base-model cross-entropy term $H_{LM}$ of the information-gain metric. We compute $s(n)$ with one frozen forward pass per sample.

We test the *marginal* relationship and not the cumulative one on purpose. Cumulative steps and $\text{DL}(n)$ both increase monotonically with $n$, so their correlation is near 1 trivially—a length artifact rather than evidence. Differencing both removes it and isolates the question of whether a *surprising* to-

*Table 12.* Full sweep of activation alignment and low-dimensional projection hyperparameters. "$\alpha$" is alignment strength, "$L$" is the number of aligned layers, and "dim" is projection dimension.

| Model | Compressed Tokens | Information Gain | Trajectory Length | PCA 99% |
|---|---|---|---|---|
| Llama-3.1-8B $\alpha = 1.0\ L = 2$ | $1532.3 \pm 280$ | $4694 \pm 341$ | $812 \pm 167$ | $45.8 \pm 5.29$ |
| Llama-3.1-8B $\alpha = 1.0\ L = 4$ | $1593.6 \pm 306.6$ | $4961 \pm 603$ | $958 \pm 207$ | $42.9 \pm 3.01$ |
| Llama-3.1-8B $\alpha = 1.0\ L = 8$ | $1633.3 \pm 275.4$ | $5130 \pm 247$ | $1146 \pm 137$ | $40 \pm 1.26$ |
| Llama-3.1-8B $\alpha = 1.0\ L = 16$ | $1487.3 \pm 255.4$ | $4686 \pm 276$ | $1146 \pm 112$ | $33.6 \pm 2.2$ |
| Llama-3.1-8B $\alpha = 1.0\ L = 24$ | $664.8 \pm 99.2$ | $2095 \pm 247$ | $570 \pm 87$ | $20.5 \pm 1.75$ |
| Llama-3.1-8B $\alpha = 1.0\ L = 32$ | $192.7 \pm 50.6$ | $614 \pm 169$ | $150 \pm 31$ | $12.6 \pm 1.69$ |
| Llama-3.1-8B dim=32 $\alpha = 1.0\ L = 8$ | $1804.8 \pm 297.8$ | $5103 \pm 748$ | $13584 \pm 3823$ | $34.7 \pm 5.37$ |
| Llama-3.1-8B dim=64 $\alpha = 1.0\ L = 8$ | $1853.3 \pm 256.2$ | $5512 \pm 456$ | $14367 \pm 3209$ | $30.8 \pm 5.27$ |
| Llama-3.1-8B dim=128 $\alpha = 1.0\ L = 8$ | $1858.9 \pm 283.6$ | $5145 \pm 1695$ | $18967 \pm 3766$ | $33.5 \pm 2.84$ |
| Llama-3.1-8B dim=256 $\alpha = 1.0\ L = 8$ | $1810.6 \pm 236$ | $5105 \pm 1693$ | $23576 \pm 5844$ | $34.1 \pm 5.5$ |
| pythia-1.4b $\alpha = 1.0\ L = 4$ | $182.7 \pm 31.1$ | $553 \pm 63$ | $83 \pm 9$ | $14.6 \pm 1.96$ |
| pythia-1.4b $\alpha = 1.0\ L = 8$ | $178.7 \pm 33.1$ | $524 \pm 92$ | $80 \pm 13$ | $14.3 \pm 1.79$ |
| pythia-1.4b $\alpha = 1.0\ L = 16$ | $155.5 \pm 35.5$ | $427 \pm 66$ | $69 \pm 16$ | $13.5 \pm 1.5$ |
| pythia-1.4b $\alpha = 1.0\ L = 20$ | $116.6 \pm 26.1$ | $307 \pm 35$ | $58 \pm 10$ | $11.9 \pm 0.83$ |
| pythia-1.4b dim=32 $\alpha = 1.0\ L = 8$ | $423.1 \pm 104.4$ | $1317 \pm 221$ | $2498 \pm 1438$ | $15.5 \pm 4.57$ |
| pythia-1.4b dim=64 $\alpha = 1.0\ L = 8$ | $447.8 \pm 81.9$ | $1398 \pm 186$ | $3309 \pm 2569$ | $16.9 \pm 4.01$ |
| pythia-1.4b dim=128 $\alpha = 1.0\ L = 8$ | $468.3 \pm 68.6$ | $1448 \pm 106$ | $4699 \pm 2414$ | $20 \pm 4.96$ |
| pythia-1.4b dim=256 $\alpha = 1.0\ L = 8$ | $498.2 \pm 92.7$ | $1555 \pm 146$ | $5396 \pm 3309$ | $19.6 \pm 5.89$ |
| SmolLM2-1.7B $\alpha = 1.0\ L = 8$ | $148.1 \pm 56.7$ | $432 \pm 157$ | $75 \pm 18$ | $11.9 \pm 1.97$ |
| SmolLM2-1.7B $\alpha = 1.0\ L = 4$ | $186.9 \pm 47.3$ | $530 \pm 104$ | $85 \pm 8$ | $13.3 \pm 1.73$ |
| SmolLM2-1.7B $\alpha = 1.0\ L = 16$ | $126.5 \pm 63.3$ | $366 \pm 155$ | $66 \pm 22$ | $11.3 \pm 2.19$ |
| SmolLM2-1.7B $\alpha = 1.0\ L = 20$ | $96.1 \pm 56.2$ | $254 \pm 114$ | $50 \pm 20$ | $8.8 \pm 3.54$ |
| SmolLM2-1.7B dim=32 $\alpha = 1.0\ L = 8$ | $292.7 \pm 113.3$ | $851 \pm 327$ | $1144 \pm 705$ | $11.1 \pm 3.99$ |
| SmolLM2-1.7B dim=64 $\alpha = 1.0\ L = 8$ | $321.4 \pm 148.3$ | $914 \pm 305$ | $1668 \pm 2014$ | $10.4 \pm 2.29$ |
| SmolLM2-1.7B dim=128 $\alpha = 1.0\ L = 8$ | $363.2 \pm 67.2$ | $1065 \pm 213$ | $2236 \pm 1198$ | $10.4 \pm 1.5$ |
| SmolLM2-1.7B dim=256 $\alpha = 1.0\ L = 8$ | $468 \pm 243.7$ | $1350 \pm 621$ | $6544 \pm 9765$ | $11.6 \pm 3.61$ |
| gemma-3-4b-pt $\alpha = 1.0\ L = 4$ | $60.3 \pm 52.7$ | $224 \pm 163$ | $60 \pm 20$ | $9.6 \pm 3.01$ |
| gemma-3-4b-pt $\alpha = 1.0\ L = 8$ | $75.1 \pm 84.9$ | $268 \pm 256$ | $65 \pm 44$ | $9.3 \pm 3.93$ |
| gemma-3-4b-pt $\alpha = 1.0\ L = 16$ | $55.8 \pm 42.8$ | $217 \pm 156$ | $57 \pm 19$ | $8.4 \pm 2.65$ |
| gemma-3-4b-pt $\alpha = 1.0\ L = 20$ | $48.3 \pm 31.4$ | $191 \pm 104$ | $55 \pm 16$ | $8.3 \pm 2.45$ |
| gemma-3-4b-pt dim=32 $\alpha = 1.0\ L = 8$ | $121.3 \pm 127.9$ | $393 \pm 317$ | $429 \pm 277$ | $9.3 \pm 3.93$ |
| gemma-3-4b-pt dim=64 $\alpha = 1.0\ L = 8$ | $327.5 \pm 224.5$ | $1082 \pm 708$ | $807 \pm 502$ | $13.4 \pm 3.58$ |
| gemma-3-4b-pt dim=128 $\alpha = 1.0\ L = 8$ | $361.3 \pm 214.4$ | $1209 \pm 672$ | $1511 \pm 801$ | $14.5 \pm 2.06$ |
| gemma-3-4b-pt dim=256 $\alpha = 1.0\ L = 8$ | $217.1 \pm 146.2$ | $719 \pm 425$ | $1091 \pm 557$ | $13.3 \pm 2.69$ |
| Qwen3-4B $\alpha = 1.0\ L = 4$ | $166.9 \pm 56.1$ | $699 \pm 193$ | $104 \pm 18$ | $13.8 \pm 1.54$ |
| Qwen3-4B $\alpha = 1.0\ L = 8$ | $197.1 \pm 74$ | $792 \pm 243$ | $118 \pm 25$ | $15.1 \pm 2.12$ |
| Qwen3-4B $\alpha = 1.0\ L = 16$ | $170.9 \pm 71.3$ | $711 \pm 265$ | $109 \pm 28$ | $13.9 \pm 2.47$ |
| Qwen3-4B $\alpha = 1.0\ L = 20$ | $168.9 \pm 39.6$ | $695 \pm 132$ | $106 \pm 12$ | $13.9 \pm 1.37$ |
| Qwen3-4B dim=32 $\alpha = 1.0\ L = 8$ | $774.9 \pm 156.4$ | $3063 \pm 556$ | $4243 \pm 1543$ | $25.9 \pm 4.48$ |
| Qwen3-4B dim=64 $\alpha = 1.0\ L = 8$ | $811.8 \pm 189.1$ | $3181 \pm 567$ | $7120 \pm 2953$ | $30.7 \pm 4.17$ |
| Qwen3-4B dim=128 $\alpha = 1.0\ L = 8$ | $870.8 \pm 160.5$ | $3400 \pm 300$ | $9040 \pm 3358$ | $31 \pm 5.42$ |
| Qwen3-4B dim=256 $\alpha = 1.0\ L = 8$ | $895.7 \pm 219.4$ | $3470 \pm 476$ | $9813 \pm 3019$ | $32.5 \pm 4.25$ |

ken is genuinely harder to cram. Because the two quantities are heavy tailed we report the rank (Spearman) correlation $\rho$, and we accompany it with three reliability checks. A 95% confidence interval comes from a cluster bootstrap that re-samples the 50 PG19 samples with replacement (the honest unit of independence, since tokens within a sample are correlated). A *partial* Spearman correlation controls for prefix position $n$, ruling out the possibility that the association is merely a shared trend with sequence depth—position itself is nearly uncorrelated with surprisal ($\rho \approx -0.1$). Finally we report the distribution of the per-sample correlation and the fraction of samples in which it is positive.

Table 16 reports these statistics for four base models spanning distinct families, tokenizers, and scales (Llama-3.1-8B, Pythia-1.4B, SmolLM2-1.7B, Gemma-3-4B), each at its main PG19 learning rate. The correlation is positive and well separated from zero for every model ($\rho$ from 0.44 to 0.59, all 95% intervals excluding 0), it holds within *every one* of the 50 samples of each run (100% positive), and—decisively—it survives controlling for position: the partial correlation $\rho \mid n$ equals or exceeds the pooled $\rho$ in all four cases (0.49–0.66). The marginal cramming cost of a token is thus governed by how much new information that token carries under the base model, not by where it sits in the

*Table 13.* Effect of low-dimensional projection size on progressive cramming. "dim" in the model column denotes the projection dimension $k$ (Section 4.2).

| Model | Compressed Tokens | Information Gain | Trajectory Length | PCA 99% |
|---|---|---|---|---|
| Llama-3.1-8B lr=0.1 | 1063.5 ± 394.4 | 3028 ± 1321 | 4861 ± 1033 | 74.4 ± 7.94 |
| Llama-3.1-8B dim=32 | 1745 ± 306 | 5312 ± 330 | 10250 ± 3256 | 35.7 ± 5.1 |
| Llama-3.1-8B dim=64 | 1834.7 ± 251 | 5694 ± 333 | 9977 ± 2641 | 32.5 ± 4.3 |
| Llama-3.1-8B dim=128 | 1834.6 ± 305.6 | 5538 ± 537 | 18713 ± 10220 | 32.8 ± 4.6 |
| Llama-3.1-8B dim=256 | 1730.8 ± 384.3 | 4870 ± 1680 | 21448 ± 5583 | 29.4 ± 3.29 |
| Llama-3.1-8B dim=512 | 1652.9 ± 404.5 | 5072 ± 799 | 25852 ± 7528 | 36.2 ± 5.88 |
| pythia-1.4b lr=0.5 | 543.8 ± 56.8 | 1694 ± 125 | 6730 ± 390 | 49 ± 3.32 |
| pythia-1.4b dim=32 | 358.2 ± 80.6 | 1137 ± 241 | 1551 ± 954 | 15.1 ± 3.01 |
| pythia-1.4b dim=64 | 392.8 ± 96.6 | 1240 ± 286 | 1920 ± 1191 | 16.7 ± 3.69 |
| pythia-1.4b dim=128 | 373 ± 81.8 | 1168 ± 240 | 1915 ± 1133 | 15.8 ± 3.22 |
| pythia-1.4b dim=256 | 375.5 ± 100.7 | 1189 ± 314 | 2135 ± 1312 | 16.5 ± 2.16 |
| pythia-1.4b dim=512 | 415.5 ± 76.3 | 1333 ± 200 | 2863 ± 1433 | 18 ± 2.49 |
| SmolLM2-1.7B lr=0.1 | 370.2 ± 113.1 | 1119 ± 350 | 1027 ± 167 | 29.2 ± 2.93 |
| SmolLM2-1.7B dim=32 | 335.8 ± 78.7 | 1007 ± 284 | 543 ± 218 | 14.2 ± 1.99 |
| SmolLM2-1.7B dim=64 | 403.2 ± 77.7 | 1195 ± 177 | 778 ± 167 | 13.8 ± 1.94 |
| SmolLM2-1.7B dim=128 | 431.9 ± 115.6 | 1252 ± 288 | 1236 ± 653 | 13.9 ± 2.51 |
| SmolLM2-1.7B dim=256 | 487.7 ± 152.7 | 1449 ± 402 | 2265 ± 1405 | 16 ± 4.47 |
| SmolLM2-1.7B dim=512 | 557.1 ± 158.8 | 1615 ± 350 | 2590 ± 1163 | 17.7 ± 3.95 |
| gemma-3-4b-pt lr=0.1 | 286.5 ± 164.6 | 949 ± 466 | 1044 ± 328 | 31.8 ± 9.21 |
| gemma-3-4b-pt dim=32 | 233.7 ± 202.3 | 670 ± 428 | 673 ± 343 | 12 ± 3.1 |
| gemma-3-4b-pt dim=64 | 214.9 ± 172.8 | 718 ± 568 | 700 ± 486 | 11.8 ± 2.71 |
| gemma-3-4b-pt dim=128 | 275.5 ± 222.6 | 860 ± 667 | 1107 ± 1265 | 13.3 ± 2.76 |
| gemma-3-4b-pt dim=256 | 252.7 ± 189.9 | 829 ± 635 | 1372 ± 886 | 13.1 ± 4.83 |
| gemma-3-4b-pt dim=512 | 217.4 ± 169.1 | 639 ± 439 | 1297 ± 975 | 12.3 ± 5.95 |

*Table 14.* Prefix-tuning attention mass, computed with the same metric as Table 26 but for learned prefix tokens instead of the compression embedding.

| Model | Prefix Token (%) | BOS Token Original (%) | Diff (%) | Correlation |
|---|---|---|---|---|
| Llama-3.2-3B | 23.24 ± 4.55 | 67.75 ± 1.18 | -44.51 ± 5.61 | 0.5641 |
| SmolLM2-135M | 26.53 ± 2.78 | 41.74 ± 6.35 | -15.21 ± 7.13 | 0.8691 |
| SmolLM2-360M | 28.07 ± 4.8 | 49.81 ± 7.67 | -21.74 ± 8.13 | 0.6066 |
| SmolLM2-1.7B | 36.43 ± 4.58 | 44.03 ± 6.73 | -7.61 ± 8.38 | 0.9102 |
| Qwen3-4B | 1.95 ± 0.22 | 50.84 ± 1.54 | -48.89 ± 1.46 | 0.0771 |

sequence.

This carries two implications. First, the procedure's cost is *information-theoretic*: predictable continuations are nearly free, while the optimizer's effort concentrates on high-surprisal tokens, so the cumulative steps to a given prefix track its description length $\mathrm{DL}(n)$ rather than its raw length. This is the per-token mechanism behind the aggregate picture in Appendix B.3, where coarsening the schedule ($\Delta > 1$) reshapes and shortens the optimization trajectory while leaving the achieved prefix length essentially unchanged—the same total information must be absorbed however it is grouped. Second, the relationship is practically useful: because $s(n)$ comes from a single frozen forward pass, the relatively difficult regions of a document can be anticipated *before* running the expensive inner optimization, which could guide step-budget allocation or early stopping.

One caveat bounds the claim. The correlation is moderate rather than deterministic ($\rho \approx 0.5$): surprisal ex-

plains a large share of the rank ordering of difficulty, but optimization-landscape effects, interactions between co-compressed tokens, and repeated structure contribute the remainder (Gemma-3-4B is the weakest and most variable across samples).

## C.3. Equally-Good Solutions Are Far Apart: Low-Dimensionality Is a Path Property

Section 5.1 reports that a single progressive-cramming trajectory occupies a low-dimensional subspace (30–100 principal components for 99% of its variance). A natural question is whether this reflects the geometry of the *set of valid solutions*—the compression embeddings that perfectly reconstruct a given prefix—or merely the geometry of one optimization *path*. We can separate the two at no additional training cost using the learning-rate sweep of Table 7. Those runs share the default random seed and initialization method, so for a given sample every learning rate starts from

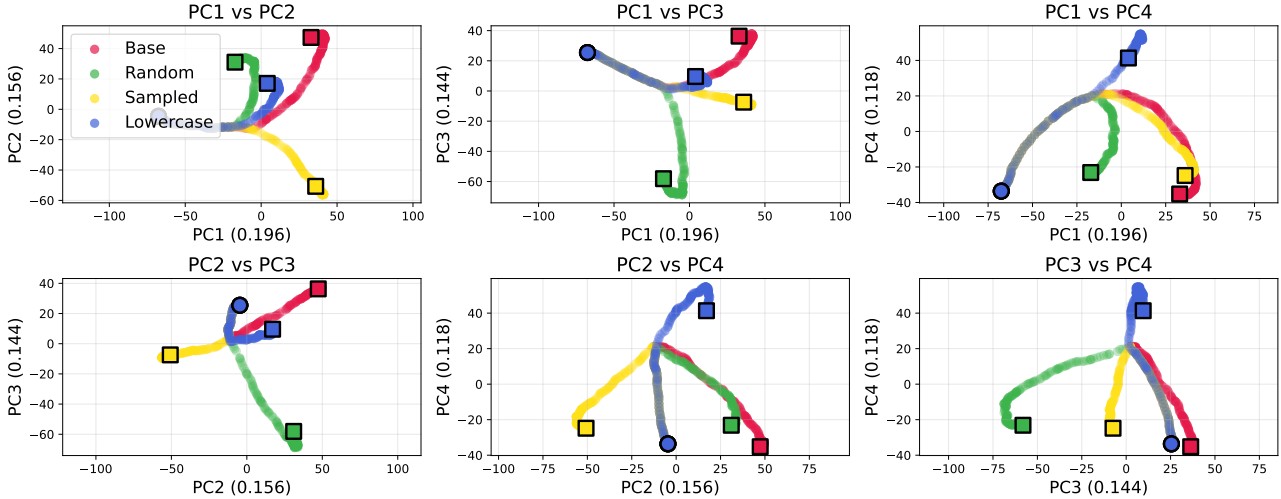

*Figure 8.* Progressive trajectories projected onto PCA components (Llama-3.1-8B, one PG19 sample). All variants share the same 64-token prefix; the suffix is either original (Base), randomly permuted (Random), greedily sampled continuation (Sampled), or lowercased (Lowercased). Circles mark the start and squares mark the end. Axis titles report the signed explained-variance fraction.

*Table 15.* Prefix-tuning reconstruction accuracy compared to progressive cramming. "Tokens" denotes the number of learned prefix tokens (or reconstructed tokens for progressive cramming).

| Experiment | Type | Tokens | Accuracy |
|---|---|---|---|
| Llama-3.2-1B | Progr. | $402 \pm 85$ | 1.0 |
| Llama-3.2-1B | Full PrefixT. | 4096 | $1 \pm 0$ |
| Llama-3.2-1B | Full PrefixT. | 8192 | $0.977 \pm 0.032$ |
| Llama-3.2-1B | Full PrefixT. | 16384 | $0.767 \pm 0.105$ |
| Llama-3.2-3B | Progr. | $902 \pm 207$ | 1.0 |
| Llama-3.2-3B | Full PrefixT. | 8192 | $1 \pm 0$ |
| Llama-3.1-8B | Progr. | $1064 \pm 394$ | 1.0 |
| Llama-3.1-8B | Full PrefixT. | 8192 | $1 \pm 0$ |
| Pythia160m | Progr. | $11 \pm 2$ | 1.0 |
| Pythia160m | Full PrefixT. | 1024 | $0.488 \pm 0.06$ |
| Pythia160m | Full PrefixT. | 2048 | $0.394 \pm 0.034$ |
| Pythia160m | Full PrefixT. | 4096 | $0.263 \pm 0.025$ |
| Pythia160m | Full PrefixT. | 8192 | $0.168 \pm 0.018$ |
| Pythia160m | Full PrefixT. | 16384 | $0.139 \pm 0.026$ |
| Pythia410m | Progr. | $102 \pm 41$ | 1.0 |
| Pythia410m | Full PrefixT. | 2048 | $1 \pm 0$ |
| Pythia410m | Full PrefixT. | 4096 | $0.964 \pm 0.071$ |
| Pythia410m | Full PrefixT. | 8192 | $0.588 \pm 0.106$ |
| Pythia410m | Full PrefixT. | 16384 | $0.418 \pm 0.158$ |
| Pythia1.4b | Progr. | $544 \pm 57$ | 1.0 |
| Pythia1.4b | Full PrefixT. | 4096 | $1 \pm 0$ |
| Pythia1.4b | Full PrefixT. | 8192 | $0.979 \pm 0.015$ |
| Pythia1.4b | Full PrefixT. | 16384 | $0.663 \pm 0.12$ |

a *byte-identical* compression embedding $e_0$ and differs only in optimizer step size. Whenever two or more learning rates drive the same prefix (sample, length) to $100\%$ reconstruction, we obtain a set of *equally valid* solutions $\{e_\eta^*\}$ reached from one common starting point.

For every such matched group we measure three things. The solutions are equally good by construction (all reach perfect reconstruction); we confirm they are also equivalent in capacity through the coefficient of variation of their information gain ("IG CV"). We measure how far apart they are by the mean pairwise distance between the $e_\eta^*$, relative to their mean displacement-from-init $\|e_\eta^* - e_0\|$ ("Sol. dist."). This ratio is scale-free, so a large value is not just an artifact of the larger steps high learning rates take. Finally we measure whether the solutions differ along shared or independent directions through the mean pairwise cosine of the displacement vectors $(e_\eta^* - e_0)$ ("Dir. cos."); the baseline for unrelated directions in $d$ dimensions is $\approx 1/\sqrt{d}$, i.e. $0.016$–$0.022$ for these hidden sizes. We pool these over all matched groups (a geometric grid of prefix lengths $\times$ samples) for each of the four families.

Table 17 shows a consistent picture. The solutions are genuinely equally good: information gain varies by only $0.9$–$1.8\%$ across learning rates at a matched prefix. Yet they are far apart—the mean cross-solution distance is $1.5$–$1.6\times$ the displacement from the shared init, meaning two equally-good solutions are *farther from each other than either is from the common starting point*. And they differ along largely independent directions: the cosines between the displacement vectors ($+0.04$ to $+0.24$) are small but positive—above the random-direction baseline—so the solutions share only a weak common component, while most of each displacement lies in directions unique to it. The learning rate, meanwhile, acts almost purely as a travel-distance knob: the mean displacement from init grows $15$–$43\times$ from the smallest to the largest learning rate ("Disp. ratio"), mirroring the order-of-magnitude growth in trajectory length and

*Table 16.* Per-token surprisal predicts the marginal number of optimization steps progressive cramming spends to absorb a token. For each base model we align, over all perfectly converged single-token stages, the marginal step cost $\Delta\text{steps}(n)$ with the frozen base model's next-token surprisal $s(n) = -\log_2 p(x_n \mid x_{<n})$, and report their Spearman rank correlation. "$N$" is the number of aligned token-level pairs (pooled over 50 PG19 samples). "Spearman $\rho$ [95% CI]" is the pooled correlation with a cluster-bootstrap interval that resamples whole samples. "Partial $\rho \mid n$" controls for prefix position, isolating the surprisal effect from any shared trend with sequence depth. "Per-sample $\bar{\rho}$" is the mean $\pm$ standard deviation of the within-sample correlations, and "% $\rho$>0" the fraction of samples with a positive correlation. The relationship is positive, reliable, and not a position artifact for every family, tokenizer, and scale tested.

| Model | $N$ | Spearman $\rho$ [95% CI] | Partial $\rho \mid n$ | Per-sample $\bar{\rho}$ | % $\rho$>0 |
|---|---|---|---|---|---|
| Llama-3.1-8B lr=0.1 | 71,778 | 0.59 [0.55, 0.63] | 0.61 | 0.57 $\pm$ 0.14 | 100% |
| Pythia-1.4B lr=0.5 | 21,419 | 0.56 [0.54, 0.59] | 0.66 | 0.57 $\pm$ 0.08 | 100% |
| SmolLM2-1.7B lr=0.1 | 16,654 | 0.53 [0.51, 0.55] | 0.63 | 0.53 $\pm$ 0.07 | 100% |
| Gemma-3-4B lr=0.1 | 10,579 | 0.44 [0.38, 0.49] | 0.49 | 0.48 $\pm$ 0.19 | 100% |

PCA-99% reported in Table 7, while information gain stays flat.

Together these reframe the low-dimensionality result. The valid-solution set for a prefix is *wide and high-dimensional*: from a single starting point, different step sizes reach equally-good solutions scattered in largely independent directions, and a higher learning rate simply reaches farther-flung members of that set. The low PCA-99% of any one trajectory (Section 5.1) is therefore a property of a single, lazily warm-started *path*—one thin slice of the solution set—not of an intrinsic low-dimensional solution manifold. This directly explains why low-dimensional projection (Section 4.2) preserves cramming quality. If valid solutions are spread densely across the embedding space, even a random rank-$k$ affine subspace still intersects the solution set, so quality is retained while the optimization geometry simplifies. It also explains why, for a model such as Llama-3.1-8B, raising the learning rate inflates trajectory length and dimensionality without improving information gain—the extra exploration finds different solutions, not better ones.

Two caveats temper the claim. First, although the scale-free distance and the direction cosines are robust to the step-size confound, these solutions still share an initialization and warm-start history, so they are correlated rather than fully independent draws; their positive direction cosines (largest for Pythia, smallest for Gemma) reflect this shared component. Second, the cleanest test—independent random restarts at a *fixed* learning rate—would remove the shared-init confound entirely; we expect it to strengthen the conclusion, since independent initializations can only increase diversity, and leave it to future work.

### C.4. Trajectory Shape: Smooth Wandering or Jumps Between Basins?

Appendix C.3 establishes that a trajectory's low dimensionality is a property of one warm-started *path*, not of an intrinsic low-dimensional solution manifold. This raises a complementary question about that same path: what does it look like *geometrically*? Is it a smooth low-dimensional curve, or a punctuated walk that dwells in a region and then

leaps to a far-apart one—the "disconnected local-minima basins" picture suggested by the variable per-token cramming cost of Appendix C.2?

**Unit and method.** For each sample we take the converged embeddings in order (one per growing-prefix stage) and analyse them in their own top-$k$ PCA subspace ($k$ = PCA 99%, the same low-dimensional anchor as Table 8). We summarise the consecutive jumps $d_k = \|e^{(k)} - e^{(k-1)}\|$ and calibrate every statistic against two nulls passed through the identical pipeline. The *smooth-curve* null fixes what an evenly traced low-dimensional curve looks like: its jump gap-ratio (max / median jump) is $\approx 1$, so a larger real value means a gappy, heavy-tailed jump distribution. The decisive *jump-shuffle* null randomly reorders the real jump vectors and re-integrates them: this holds the jump-*size* distribution exactly fixed and randomises only their order, so the lag-1 autocorrelation of jump magnitudes $r_1$ is $\approx 0$ by construction. A real $r_1$ above the shuffle band therefore means small and large jumps are bunched *in order*—runs of small steps punctuated by occasional large ones—which is precisely the dwell-then-leap signature of moving between discrete basins, and cannot be reproduced by the jump-size distribution alone. We call a sample "dwelling" when its $r_1$ exceeds the shuffle-null 95th percentile; "Dwelling %" is the fraction of samples that qualify. The analysis reads only already-saved trajectories and adds no training cost.

**Trajectories are gappy at every scale, but only dwell at large scale.** Two facts hold for every run (Table 18): the trajectories are low-dimensional (PCA 99% of 13–83) yet decisively *not* smooth curves (jump gap-ratio 7–22 versus a smooth-curve null of $\approx$ 1.1–1.3). What changes with scale is whether that gappiness reflects *genuine dwelling*. At the two smallest models the jump autocorrelation sits at or below the shuffle null ($r_1 = +0.00$ at 135M and $-0.04$ at 360M; only 10–12% of samples dwelling): the gaps are fully explained by a heavy-tailed distribution of jump *sizes*, i.e. heavy-tailed wandering rather than discrete basins. Genuine dwelling then switches on with scale—$r_1 = +0.11$ at 1.7B (60% of samples dwelling)—and saturates at Llama-

*Table 17.* Equally-good cramming solutions are far apart and point in largely independent directions. Using the shared-initialization learning-rate sweep of Table 7, we pool over all prefix (sample, length) groups that two or more learning rates drive to perfect reconstruction, and report: "$N$" the number of such groups (over 10 PG19 samples); "IG CV" the coefficient of variation of information gain across the learning rates (equal quality $\Rightarrow$ small); "Sol. dist." the mean cross-solution Euclidean distance in units of the mean displacement-from-init (median [IQR]; $>1$ means solutions are farther from each other than from the shared start); "Dir. cos." the mean pairwise cosine of the solutions' displacement vectors, each measured from the shared init (random baseline $\approx 1/\sqrt{d} = 0.016$–$0.022$); and "Disp. ratio" the mean displacement-from-init at the largest learning rate over that at the smallest. Equally-good solutions are far apart and point in largely independent directions across every family, indicating a wide, high-dimensional valid-solution set.

| Model | $N$ | IG CV | Sol. dist. | Dir. cos. | Disp. ratio |
|---|---|---|---|---|---|
| Llama-3.1-8B | 224 | 0.9% | 1.49 [1.43, 1.53] | +0.18 | 15× |
| Pythia-1.4B | 238 | 1.8% | 1.58 [1.55, 1.62] | +0.24 | 43× |
| SmolLM2-1.7B | 239 | 1.6% | 1.61 [1.58, 1.64] | +0.10 | 27× |
| Gemma-3-4B | 198 | 1.8% | 1.63 [1.61, 1.64] | +0.04 | 19× |

3.1-8B, where $r_1 = +0.44$ and *every* sample dwells. The strong "disconnected far-apart basins" reading of the trajectory is thus *false at small scale ($\leq 360M$) and emerges, then saturates, at larger scale*, with basins that grow increasingly well separated (mean inter- vs. intra-cluster distance rising from $\approx 10\times$ at 135M to $\approx 73\times$ at 8B).

**Learning rate tunes the dwelling, non-monotonically.** Holding the model fixed (SmolLM2-1.7B; Table 19), the dwelling signal peaks at the *middle* step size: $r_1 = +0.11, +0.19, +0.15$ (60%, 92%, 78% dwelling) at learning rates 0.1, 0.5, 1.0. Too-small steps barely separate successive basins, while too-large steps blur the dwell-and-leap structure; the intermediate rate traces the cleanest staircase. Consistent with the displacement-knob role of the learning rate in Appendix C.3, larger rates also inflate PCA 99% and the overall gappiness.

**Effort, not distance: jumps in optimization steps.** The jumps above are Euclidean distances in embedding space. Besides depending on the model's hidden size (so they are not comparable across families), a Euclidean metric treats that space as flat and isotropic, whereas the frozen model reads each embedding through a deeply non-linear function—so equal Euclidean steps need not be equally consequential. A more faithful measure of how far the optimizer actually travelled between two converged points may be how much *work* the move cost: we redefine the jump as the *number of optimization steps* spent to absorb that token (the marginal per-token step count of Appendix C.2), a model-agnostic, effort-based unit that implicitly follows the curved loss geometry the optimizer descends rather than straight-line distance, and run the identical jump-shuffle test on this scalar series (the step-based columns of Tables 18 and 19). Two things change. First, ordered dwelling is now present at *every* scale: the step-series autocorrelation is $r_1 \approx +0.16$ already at 135M (versus $+0.00$ in embedding space) and rises only mildly to $+0.29$ at Llama-3.1-8B, so even small models alternate runs of cheap tokens with occasional expensive ones—bursty difficulty—even where their *spatial* jumps are

not bunched. Second, although this autocorrelation is nearly flat across scale, the dwelling *fraction* still climbs from 38% to 100%: that gap is the signature of a statistical-power effect, since a longer trajectory has more jumps and the *same* modest autocorrelation clears the per-sample shuffle band in more samples. The effort-based view thus adds a caveat to the distance-based scale trend—because trajectory length grows steeply with the compression horizon ($\sim 38$ stages at 135M to $\sim 1438$ at Llama-3.1-8B), the growth of the dwelling *fraction* with scale is driven largely by statistical power, not by stronger per-step ordering.

**How much of the low dimensionality is trivial? A random-walk null.** A low PCA 99% (the trajectory dimensionality in Table 18) need not imply directional structure: *any* running sum of steps is autocorrelated in time, so even a random walk occupies far fewer dimensions than it has steps. To separate this generic cumulative-sum effect from trajectory-specific structure, we compare each trajectory's PCA 99% against an *i.i.d. random-walk null* that re-integrates the sample's own jump *magnitudes*, randomly permuted, along *random isotropic directions*—matching the jump-size distribution exactly while destroying both temporal order and orientation. The verdict is scale-dependent. At $\leq 1.7B$ the real PCA 99% sits slightly *below* the null (13.1 vs. 14.7 at 135M; 15.3 vs. 18.3 at 360M; 33.2 vs. 40.6 at 1.7B; ratios 0.82–0.89), so much of the small-model low dimensionality is compatible with generic random-walk geometry plus mild additional directional smoothing—an honest caveat to "surprisingly low-dimensional" at small scale. At Llama-3.1-8B the comparison reverses: the real PCA 99% (82.8) is $\approx 1.6\times$ the null (51.3), i.e. the trajectory spans *more* dimensions than a magnitude-matched random walk. This means the 8B trajectory is not merely a cumulative-sum artifact: it has additional organized directional spread, even though the absolute count (83 of 4096) remains small. That pattern is consistent with the dwell-and-leap reading from the jump-shuffle test—ordered movement between regions can add fresh directions beyond the diffusive baseline—but the PCA-rank null alone should be read as corroborative

rather than causal evidence for basins. A simpler "a few big jumps explain it" story also fails—at 1.7B the subspace spanned by the largest-magnitude jumps captures only 52% of trajectory variance, and 225 of 334 jumps are needed to reach 99%. Neither a trivial random walk nor a handful of dominant moves fully accounts for the geometry; the trajectory retains low absolute rank while developing additional ordered directional structure at the largest scale.

**Relation to the solution set.** The two analyses are complementary cuts of the same object. Appendix C.3 shows the *valid-solution set* for a prefix is wide and high-dimensional and that any one trajectory is a thin low-dimensional slice of it; here we add that the slice itself is not a smooth curve but, at sufficient scale, a punctuated walk through discrete, well-separated minima. Low dimensionality and discreteness coexist. Two caveats bound the claim. The smooth-curve null alone is a weak bar—every run clears it—so it would misleadingly label all models "clustered"; it is the jump-shuffle autocorrelation that separates genuine dwelling from heavy-tailed wandering, and the component-count and inter/intra-distance figures are corroborative rather than primary. All runs use 50 PG19 samples (49 at 360M, where one sample had too few stages to analyse). The small-scale end is not perfectly ordered—360M sits marginally *below* 135M—but both are statistically indistinguishable from zero dwelling, so the qualitative split (no dwelling at $\leq$360M, clear dwelling at $\geq$1.7B) is unaffected.

# D. Reconstruction Fidelity and Downstream Capability

## D.1. Analysis of the Reconstruction Accuracy

As noted in Section 3.2, a 99% token-level accuracy threshold is insufficient to guarantee correct reconstruction when evaluation is performed without *teacher forcing* (i.e., when generated tokens are fed back into the prefix rather than replacing them with the ground-truth tokens). Table 20 reports per-position mismatch under both regimes: under teacher forcing the error is concentrated in the first few positions (indices 0–1, tapering thereafter), whereas under greedy decoding these early mistakes compound, keeping the per-position mismatch near 100% for the rest of the sequence and leading to divergent generations.

## D.2. Effect of Imperfect Reconstruction on HellaSwag and ARC

The downstream evaluation in Section 6 retains benchmark instances whose prefixes are not perfectly reconstructed, which raises the question of whether the observed capability drop is caused by the compression embedding itself disrupting reasoning, or simply by failed reconstruction corrupting

the prefix. To disentangle the effect of imperfect reconstruction from the effect of the compression embedding itself, we emphasize the difference between autoregressive generation and multiple-choice benchmarks such as HellaSwag and ARC. In greedy generation, even small reconstruction errors can lead to large deviations in the output due to error accumulation, especially when errors occur early in the sequence (Section 3.2).

In contrast, HellaSwag and ARC operate in a non-generative (scoring) regime: the model evaluates fixed context–continuation pairs without regenerating the prefix. As a result, these benchmarks are significantly less sensitive to small reconstruction imperfections, since no sequential error propagation occurs.

The near-identical token-normalized accuracies for SmolLM2-1.7B (Table 21) indicate that imperfect reconstruction has minimal impact on benchmark performance. This confirms that the observed degradation is driven primarily by the compression embedding itself, rather than by reconstruction errors.

### D.2.1. Likelihood-Scoring Strategies

We further localize the degradation reported in Section 6 by varying which token logits enter the likelihood score on SmolLM2-1.7B, over perfectly reconstructed instances (Table 22). Replacing the prefix with its compression embedding while scoring the full context plus the candidate ending drops HellaSwag from 52.4% to 36.5% and ARC-Challenge from 36.7% to 28.6%. Scoring only the answer-ending tokens – which do not require reading the compressed prefix – recovers part of the gap (HellaSwag 40.7%), and replacing the context entirely with the embedding ("compression only") is worst (34.6% / 22.9%). Across every scoring strategy the compressed conditions remain well below baseline, reinforcing that faithful reconstruction does not translate into usable downstream semantics.

## D.3. Generative Benchmark Evaluation (5-shot MMLU)

The downstream evaluation in Section 6 relies on HellaSwag and ARC, which probe a limited slice of downstream behavior. To broaden this evaluation, we additionally evaluate on 5-shot MMLU (512 samples) under three compression modes: *few_shot* (only the few-shot examples are compressed), *full_prefix* (both the few-shot examples and the question are compressed), and *random* (a random embedding replaces the compression embedding as a control).

The results (Table 23) are consistent with our earlier findings. When only the few-shot tokens are compressed, all models show accuracy degradation (ranging from $-2\%$ for Gemma-3-4B to $-15\%$ for Llama-3.1-8B), confirming that the model retains generative capability when uncompressed

*Table 18.* Trajectory shape across model scale (progressive cramming, PG19, sequence length 4096, learning rate 0.1), comparing two definitions of the jump between consecutive converged points. *Euclidean-based*: the L2 distance in the trajectory's top-$k$ PCA subspace ($k$ = PCA 99%). *Step-based*: the marginal number of optimization steps spent on that token. For each we report a gap-ratio (max / median jump; the Euclidean group also gives the smooth-curve-null mean in parentheses, $\approx 1$), the lag-1 jump-magnitude autocorrelation $r_1$ (jump-shuffle null in parentheses, $\approx 0$), and "Dwelling %", the fraction of samples whose $r_1$ beats the shuffle-null 95th percentile. "Stages" is the mean converged points per sample and "PCA 99%" the mean trajectory dimensionality. Euclidean dwelling emerges only with scale, whereas step (effort) dwelling is present at every scale; the rising step Dwelling % tracks trajectory length—a statistical-power effect.

| Model | Stages | PCA99% | Euclidean-based | | | Step-based | | |
|---|---|---|---|---|---|---|---|---|
| | | | Gap-ratio | $r_1$ (shuffle) | Dwelling % | Gap-ratio | $r_1$ (shuffle) | Dwelling % |
| SmolLM2-135M | 38 | 13 | 7.2 (1.3) | +0.00 (-0.04) | 10% | 31.2 | +0.16 (-0.03) | 38% |
| SmolLM2-360M | 53 | 15 | 8.2 (1.3) | -0.04 (-0.02) | 12% | 45.8 | +0.17 (-0.02) | 39% |
| SmolLM2-1.7B | 335 | 33 | 17.5 (1.2) | +0.11 (+0.00) | 60% | 118.8 | +0.28 (+0.00) | 88% |
| Llama-3.1-8B | 1438 | 83 | 21.5 (1.1) | +0.44 (+0.00) | 100% | 219.6 | +0.29 (+0.00) | 100% |

*Table 19.* Learning-rate sweep of trajectory shape on SmolLM2-1.7B (50-sample runs; columns as in Table 18). The Euclidean dwelling signal $r_1$ is non-monotonic in the learning rate, peaking at the intermediate rate 0.5 (92% of samples dwelling) and weakening at both 0.1 and 1.0.

| Model | Stages | PCA99% | Euclidean-based | | | Step-based | | |
|---|---|---|---|---|---|---|---|---|
| | | | Gap-ratio | $r_1$ (shuffle) | Dwelling % | Gap-ratio | $r_1$ (shuffle) | Dwelling % |
| SLM2-1.7B lr=0.1 | 335 | 33 | 17.5 (1.2) | +0.11 (+0.00) | 60% | 118.8 | +0.28 (+0.00) | 88% |
| SLM2-1.7B lr=0.5 | 455 | 72 | 15.5 (1.1) | +0.19 (+0.00) | 92% | 71.0 | +0.21 (+0.00) | 94% |
| SLM2-1.7B lr=1.0 | 500 | 100 | 23.0 (1.1) | +0.15 (+0.00) | 78% | 81.5 | +0.16 (+0.00) | 82% |

*Table 20.* Compression reconstruction accuracy under non-ideal training convergence (50 PG19 samples per setting). "TF conv." is mean teacher-forcing accuracy over all compressed samples; "Greedy conv." is mean full-sequence match rate when generating autoregressively from the compression embedding. We then report the fraction of samples with a token mismatch at indices 0, 1, and 2 under two decoding regimes: greedy autoregressive decoding (errors compound) and teacher forcing (each position scored against the ground-truth prefix). Variants include ignoring the BOS loss during training (no BOS) and upweighting the loss of the first two tokens by a factor of three (2 leading).

| Model | Tokens | Setup | TF conv. | Greedy conv. | Greedy mismatch (%) | | | TF mismatch (%) | | |
|---|---|---|---|---|---|---|---|---|---|---|
| | | | | | @0 | @1 | @2 | @0 | @1 | @2 |
| Llama-3.2-1B | 512 | common | 0.9904 | 0.0019 | 86.0% | 100.0% | 100.0% | 88.0% | 100.0% | 28.0% |
| Llama-3.2-1B | 512 | no BOS | 0.9887 | 0.0380 | 56.0% | 72.0% | 82.0% | 56.0% | 34.0% | 16.0% |
| Llama-3.2-1B | 512 | 2 leading | 0.9903 | 0.0045 | 0.0% | 100.0% | 100.0% | 0.0% | 100.0% | 30.0% |
| Llama-3.2-3B | 1024 | common | 0.9902 | 0.0019 | 100.0% | 100.0% | 98.0% | 100.0% | 98.0% | 36.0% |
| Llama-3.2-3B | 1024 | no BOS | 0.9821 | 0.0150 | 80.0% | 85.0% | 90.0% | 80.0% | 62.0% | 48.0% |
| Llama-3.2-3B | 1024 | 2 leading | 0.9904 | 0.0123 | 10.0% | 98.0% | 100.0% | 10.0% | 98.0% | 42.0% |
| Llama-3.1-8B | 1568 | common | 0.9907 | 0.0041 | 100.0% | 100.0% | 100.0% | 100.0% | 100.0% | 60.0% |
| Llama-3.1-8B | 1568 | no BOS | 0.9798 | 0.0044 | 84.0% | 92.0% | 96.0% | 84.0% | 68.0% | 46.0% |
| Llama-3.1-8B | 1568 | 2 leading | 0.9908 | 0.0152 | 28.0% | 96.0% | 98.0% | 28.0% | 96.0% | 54.0% |

*Table 21.* Token-normalized accuracy (%) for HellaSwag, ARC-Easy, and ARC-Challenge on SmolLM2-1.7B. Values are reported for perfectly reconstructed examples and for all examples.

| Subset | HellaSwag | ARC-Easy | ARC-Challenge |
|---|---|---|---|
| *Perfect only* | 36.05 | 55.39 | 28.86 |
| *All samples* | 36.14 | 55.21 | 28.44 |

tokens follow the compression embedding. In contrast, *full_prefix* compression—where the compression embedding compresses the full prefix—leads to near-complete failure: for Llama-3.1-8B, SmolLM2-1.7B, and Gemma-3-4B accuracy drops to 0% with no valid (parseable) answers, while

the smaller Pythia-1.4B retains only partial output validity (24.8%) at near-chance accuracy (4.83%). In every case the generative process collapses. Notably, the *random* baseline causes only minor degradation (at most $-4.3\%$, and slightly *positive* for Gemma-3-4B), nowhere near the catastrophic collapse under *full_prefix*; this shows that the mere presence of an out-of-distribution token is not sufficient to explain the failure—it is specifically the *optimized* compression embedding that disrupts downstream computation.

These results on a generative benchmark with longer context reinforce the conclusions drawn from HellaSwag and ARC, and demonstrate that the downstream collapse generalizes

*Table 22.* Token-normalized accuracy (%) on HellaSwag, ARC-Easy (ARC-E), and ARC-Challenge (ARC-C) for SmolLM2-1.7B under different likelihood-scoring strategies, over perfectly reconstructed instances. *Baseline* uses the original prefix; *Compression* replaces the prefix with its crammed embedding; *compression only* drops the prefix and keeps only the embedding. "context + ending" scores all tokens, while "ending only" scores just the candidate-continuation tokens.

| Scoring setup | HellaSwag | ARC-E | ARC-C |
|---|---|---|---|
| **Baseline** | | | |
| (context + ending) | 52.41 | 68.72 | 36.66 |
| (ending only) | 53.30 | 53.88 | 40.29 |
| **Compression** | | | |
| (context + ending) | 36.47 | 55.87 | 28.62 |
| (ending only) | 40.70 | 41.63 | 31.36 |
| (compression only) | 34.57 | 30.92 | 22.94 |

beyond multiple-choice likelihood-based evaluation.

### D.4. Generalization to a Second Dataset (Fanfics)

The experiments in the main text evaluate compression on PG-19. To verify that our findings are not specific to a single corpus, we reproduce the Fanfics data-preparation pipeline of Kuratov et al. (2025) (the `LarryLovestein/fanfics_1k` Hugging Face dataset) and rerun progressive cramming on 50 samples across four model families (Table 24). For each family we report baseline progressive cramming, low-dimensional projection only ("dim"), activation alignment only ($\alpha$, $L$), and their combination.

The key findings are consistent with those observed on PG-19. Information gain follows the same patterns across configurations: low-dimensional projection substantially increases both the number of compressed tokens and information gain. The low-dimensional structure of compression trajectories also generalizes to Fanfics data: while the PCA-99% values are somewhat larger than those observed on PG-19—likely reflecting the greater distributional diversity of fan-fiction text—they remain at least an order of magnitude below the embedding dimensionality.

### D.5. Free-Generation Quality (LLM-as-Judge)

The downstream evaluations above probe fixed-choice tasks (multiple choice, 5-shot MMLU). To assess open-ended generation *beyond* the compressed window, we run a small LLM-as-judge study on SmolLM2-1.7B. For each of 16 PG-19 samples we compress a 32-token prefix into a single embedding and then autoregressively generate 64 tokens conditioned only on that embedding: the first $\sim$32 tokens reconstruct the prefix and the remainder is a free continuation past the compressed span. We compare two training objectives for the embedding: the standard cross-entropy

reconstruction loss (*Common*) and a hybrid loss that adds cosine activation alignment (*Hybrid*; $\alpha = 1$, $L = 5$; Appendix B.5).

An instruction-tuned judge (`grok-4.1-fast`, temperature 0, structured binary output) rates each output on five criteria: adequacy of the restored prefix, and coherence, relevance, fluency, and style consistency of the continuation. An independent judge (`gemini-2.5-flash`) yields the same conclusions, indicating the verdicts are not an artifact of the judge model.

Table 25 reports the fraction of samples passing each criterion. Prefix restoration is identical for both objectives (0.938), confirming matched reconstruction fidelity. On the free continuation, however, Hybrid is consistently better: coherence $0.500 \to 0.625$, relevance $0.625 \to 0.750$, and fluency $0.562 \to 0.625$, with style consistency unchanged (0.625). At equal reconstruction fidelity, activation alignment thus yields an embedding that carries more usable semantics for generation beyond the compressed window, reinforcing the paper's theme that reconstruction accuracy alone does not capture a representation's downstream value.

These results are a qualitative complement rather than a benchmark: the sample is small (16), the criteria are binary, and we use a single model and compression length. The study is also restricted to embeddings that reach near-perfect reconstruction; under imperfect reconstruction the free continuation degrades, consistent with the autoregressive collapse documented in Appendix D.1.

## E. Mechanism: Attention and Causality

### E.1. Attention Concentration

To probe the mechanism behind perfect reconstruction, we measure whether the compression embedding becomes an *attention sink* that captures a disproportionate fraction of attention, analogous to (and potentially competing with) the BOS token. We quantify how much attention mass other positions allocate to the first position (position 0), which corresponds to the compression embedding in crammed inputs or the BOS token in the uncompressed baseline. Let $\mathbf{A}_l \in [0,1]^{S \times S}$ denote the attention matrix at layer $l$ after averaging over heads. For a given prefix length $s \leq S$, we define the attention mass on position 0 as

$$m_l(s) = \frac{1}{s-1} \sum_{q=1}^{s-1} \mathbf{A}_l(q, 0) \quad (8)$$

which excludes self-attention ($q = 0$) and averages over query positions. We report $100 \cdot m_l(s)$ as a percentage, and average over a set of target prefix lengths $\mathcal{S}$:

$$\bar{m}_l = \frac{1}{|\mathcal{S}|} \sum_{s \in \mathcal{S}} m_l(s) \quad (9)$$

*Table 23.* 5-shot MMLU accuracy on 512 samples with a compression embedding in context. **CompMode = few_shot**: only the few-shot tokens were compressed with cramming. **CompMode = full_prefix**: few-shot tokens and question tokens were compressed. **CompMode = random**: a random embedding was placed instead of the compression embedding. **Baseline Acc**: accuracy of the base model without a compression embedding. **Compressed Acc**: accuracy with a compression embedding in context (regular tokens placed after the compression embedding). **Valid** columns report the percentage of valid generated answers (answers parsed successfully).

| Model | CompMode | Baseline Acc | Compressed Acc | Acc Diff | Baseline Valid | Compressed Valid |
|---|---|---|---|---|---|---|
| Llama-3.1-8B | few_shot | 63.22% | 48.14% | -15.08% | 100.00% | 100.00% |
| Llama-3.1-8B | full_prefix | 63.22% | 0.00% | -63.22% | 100.00% | 0.00% |
| Llama-3.1-8B | random | 63.22% | 62.71% | -0.51% | 100.00% | 100.00% |
| SmolLM2-1.7B | few_shot | 50.98% | 45.91% | -5.07% | 100.00% | 100.00% |
| SmolLM2-1.7B | full_prefix | 50.98% | 0.00% | -50.98% | 100.00% | 0.00% |
| SmolLM2-1.7B | random | 50.98% | 46.66% | -4.32% | 100.00% | 100.00% |
| gemma-3-4b-pt | few_shot | 58.13% | 56.07% | -2.05% | 100.00% | 100.00% |
| gemma-3-4b-pt | full_prefix | 58.13% | 0.00% | -58.13% | 100.00% | 0.00% |
| gemma-3-4b-pt | random | 58.13% | 59.18% | 1.05% | 100.00% | 100.00% |
| pythia-1.4b | few_shot | 28.15% | 21.27% | -6.88% | 100.00% | 95.70% |
| pythia-1.4b | full_prefix | 28.15% | 4.83% | -23.33% | 100.00% | 24.80% |
| pythia-1.4b | random | 28.15% | 27.01% | -1.14% | 100.00% | 100.00% |

*Table 24.* Progressive cramming variants and key metrics across model families on **50 samples** of the **Fanfics** dataset following Kuratov et al. (2025) (`LarryLovestein/fanfics_1k`). For each family we report baseline progressive cramming, low-dimensional projection only ("dim"), activation alignment only ($\alpha$, $L$), and their combination.

| Model | Compressed Tokens | Information Gain | Trajectory Length | PCA 99% |
|---|---|---|---|---|
| pythia-1.4b lr=0.5 | $564.5 \pm 7.5$ | $1255 \pm 309$ | $6331 \pm 739$ | $46.12 \pm 4.17$ |
| pythia-1.4b lr=0.5 $\alpha = 1.0$ $L = 8$ | $517 \pm 26$ | $1176 \pm 295$ | $6180 \pm 598$ | $43.06 \pm 3.49$ |
| pythia-1.4b dim=256 lr=0.5 | $619.5 \pm 12.5$ | $2083 \pm 258$ | $510014 \pm 92005$ | $124.5 \pm 14.23$ |
| pythia-1.4b dim=256 lr=0.5 $\alpha = 1.0$ $L = 8$ | $540 \pm 54$ | $1693 \pm 181$ | $494306 \pm 58027$ | $128.8 \pm 12.6$ |
| SmolLM2-1.7B lr=0.1 | $486 \pm 45$ | $710 \pm 666$ | $1038 \pm 139$ | $28.72 \pm 4.68$ |
| SmolLM2-1.7B lr=0.1 $\alpha = 1.0$ $L = 8$ | $337.5 \pm 19.5$ | $640 \pm 271$ | $878 \pm 137$ | $22.76 \pm 4.33$ |
| SmolLM2-1.7B dim=256 lr=0.1 | $1050 \pm 72$ | $3318 \pm 659$ | $321418 \pm 59468$ | $95.14 \pm 16.53$ |
| SmolLM2-1.7B dim=256 lr=0.1 $\alpha = 1.0$ $L = 8$ | $968 \pm 1$ | $3396 \pm 342$ | $305391 \pm 53905$ | $71.44 \pm 12.46$ |
| gemma-3-4b-pt lr=0.1 | $416.4 \pm 22.3$ | $-118 \pm 1241$ | $870 \pm 274$ | $24.22 \pm 10.07$ |
| gemma-3-4b-pt lr=0.1 $\alpha = 1.0$ $L = 8$ | $410.4 \pm 43.7$ | $437 \pm 509$ | $868 \pm 305$ | $23.02 \pm 9.62$ |
| gemma-3-4b-pt dim=32 lr=0.1 | $907.8 \pm 90.8$ | $2979 \pm 801$ | $84162 \pm 15369$ | $39.44 \pm 3.59$ |
| gemma-3-4b-pt dim=32 lr=0.1 $\alpha = 1.0$ $L = 8$ | $962 \pm 31.3$ | $2667 \pm 1248$ | $111260 \pm 23499$ | $43.86 \pm 3.81$ |

*Table 25.* LLM-as-judge evaluation of free continuations generated from a single compression embedding (SmolLM2-1.7B, 16 PG-19 samples, 32-token compressed prefix, 64-token generation). Values are the fraction of samples for which the judge (`grok-4.1-fast`) returns `True` on each binary criterion. *Common*: cross-entropy reconstruction; *Hybrid*: with cosine activation alignment ($\alpha = 1$, $L = 5$).

| Criterion | Common | Hybrid | $\Delta$ |
|---|---|---|---|
| Prefix restored adequately | 0.938 | 0.938 | 0.000 |
| Continuation coherent | 0.500 | 0.625 | +0.125 |
| Continuation relevant | 0.625 | 0.750 | +0.125 |
| Continuation fluent | 0.562 | 0.625 | +0.062 |
| Style consistent | 0.625 | 0.625 | 0.000 |

Table 26 shows that this attention concentration is highly model-dependent when aggregated over layers. SmolLM2 and Pythia exhibit strong correlation between the layer-wise attention-mass profiles for the BOS token and the compression embedding; in these families the compression embedding attracts substantial attention mass in the same layers where BOS is an attention sink, and in Pythia it can match or exceed BOS attention on average. In contrast, Llama and Gemma show much lower compression-embedding attention mass than BOS on average and weak (or negative) correlation, suggesting that the concentration is confined to specific layers rather than uniformly matching the BOS pattern.

Motivated by this similarity, we test whether including both a BOS token and a compression embedding in the prompt creates interference between two attention sinks. We therefore run ablations that remove the BOS token and recompute progressive cramming metrics (Table 27). Removing BOS increases reconstructed length and information gain (and reduces compression-embedding attention mass) for Llama-3.1-8B, but has little effect on the other model families.

This attention concentration is correlational: the causal analysis in Section 7 shows it is a *symptom* of cramming rather than the cause of downstream degradation.

### E.2. Attention Knockout Across Model Families

To test whether the causal picture in Section 7 is specific to Llama-3.1-8B, we repeat the identical attention-knockout protocols on two further model families: Pythia-1.4B (GPT-NeoX, $L = 24$ layers) and SmolLM2-1.7B ($L = 24$ layers). All settings match the main text (HellaSwag, 512 samples, a single compression embedding, 1000 optimization steps, per-model compression learning rate). The results are consistent with Llama-3.1-8B (Figures 9 and 10): early-layer knockout sharply degrades teacher-forced reconstruction while leaving downstream accuracy at or above the crammed baseline; the late layers that carry the most attention mass have negligible causal effect on either metric; and the forward/reverse cumulative protocols exhibit the same asymmetry that localizes the downstream collapse to the early layers.

### E.3. On Gemma's Lower Information Gain

Gemma 3 reaches lower information gain at shorter reconstructed lengths than Llama and Pythia. One candidate explanation is logit softcapping (Team et al., 2024), which bounds logit magnitudes and limits how peaked the model's next-token distributions can become: because cramming drives the model toward high-confidence, token-specific predictions over long horizons, capped logits would reduce the separability between the correct token and its close alternatives and damp gradients in the high-confidence regime. This is consistent with Gemma's lower information gain and earlier saturation, but we do not isolate softcapping from other architectural and training-time differences, and leave confirming the mechanism to future work.

## F. Compression Capacity: Depth, Width, and Recovery

### F.1. Transformer-Depth Ablation

The main text (Section 8, Table 5) summarises the headline depth–size result—compression capacity rises with both retained depth and model size—as a compact pivot of compressed-token counts. Table 28 here gives the full per-metric breakdown of that sweep (the *first-only* scheme: keep the first $N$ decoder layers, finetune to repair the cut, then measure compression, across five model families ordered by size). The rest of this appendix covers how a checkpoint is truncated, why the recovery finetune matters, and two alternative truncation schemes that confirm the same monotone-in-depth trend.

**Truncation and recovery.** For a target of $N$ retained layers we keep the chosen decoder layers, preserve the input embedding and output projection, renumber the layer indices, and update `num_hidden_layers` so each truncated model is a self-contained checkpoint. Truncation leaves the retained layers stitched across a gap they were never trained to bridge, so a truncated checkpoint is a degraded—but not retrained—language model. To separate "capacity lost to fewer layers" from "capacity lost to the un-adapted stitch," we *finetune* each truncated checkpoint with a standard next-token objective (fineweb-edu, 5k steps at sequence length 1024, effective batch $\approx$256k tokens per step, AdamW, cosine-with-min-lr schedule, peak learning rate $10^{-3}$, 500 warmup steps, bf16) before measuring compression—the same recipe as the model-width ablation (Appendix F.3). The first-only rows of Table 28 are all recovered this way; the larger Llama-3.1-8B instead uses a gentler 15k-step / $3 \times 10^{-4}$ / 1000-warmup finetune, because at the common $10^{-3}$ rate its truncations are prone to a transient warmup loss spike that leaves some checkpoints in a degraded basin. Recovery matters: for SmolLM2-1.7B at $\{2, 4, 8, 16\}$ retained layers (first+last scheme) finetuning lifts the compressed-token count from 76.9/270.9/240.8/272.5 (as-is) to 233.8/453.3/428.6/335.3, recovering—and at 4 layers exceeding—the full 24-layer model (335.1).

**Alternative truncation schemes.** The main-text table keeps the *first* $N$ layers; two alternatives confirm the depth trend is not an artifact of *which* layers are kept. Keeping the first $N$ *and* last $N$ layers (first+last), or only the last $N$ layers (last-only), both leave compression rising monotonically with retained depth: SmolLM2-1.7B last-only gives 88.8/176.0/211.7/351.5 at 1/2/4/8 layers, and Llama-3.1-8B last-only gives 56.3/116.9/330.3 at 1/2/4. Table 29 reports these alternative-scheme rows, together with the un-finetuned first+last checkpoints used for the recovery comparison above.

### F.2. Initialization Ablation

To identify which pretrained components are load-bearing for progressive cramming, we ablate *initialization* on SmolLM2-1.7B: starting from the pretrained model, we randomly re-initialize exactly one component and keep the rest pretrained. We consider three variants: (i) *random transformer layers*, where the decoder layers are re-initialized while the input embedding and (tied) output projection are preserved; (ii) *random LM head*, where only the output projection is re-initialized; and (iii) *random input embeddings*, where only the input token-embedding matrix is re-initialized. Because SmolLM2-1.7B ties its input and output embeddings, for variants (ii) and (iii) we first *untie* them - so that exactly one matrix is randomized while the other retains its pretrained values.

*Table 26.* Attention-concentration summary. "Compression Embedding" is attention mass to position 0 for the compression embedding in the crammed input; "BOS Token" is attention mass to position 0 for the uncompressed baseline; "Diff" is (Compression−BOS). "Correlation" is the Pearson correlation between per-layer attention-mass profiles. Ŗ stands for removed BOS token for cramming compression column.

| Model | Compression Embedding (%) | BOS Token Original (%) | Diff (%) | Correlation |
|---|---|---|---|---|
| Llama-3.2-1B lr=0.1 | 20.71 ± 5.74 | 67.37 ± 1.41 | -46.66 ± 6.5 | 0.2884 |
| Llama-3.2-3B lr=0.1 | 15.97 ± 1.15 | 70 ± 1.14 | -54.03 ± 1.01 | -0.1907 |
| Llama-3.1-8B lr=0.1 | 38.07 ± 17.98 | 70.56 ± 1.26 | -32.49 ± 18.96 | 0.4104 |
| Llama-3.1-8B Ŗ lr=0.1 | 18.93 ± 10.43 | 70.05 ± 1.29 | -51.12 ± 10.33 | 0.4688 |
| pythia-160m lr=0.5 | 50.88 ± 16.11 | 36.15 ± 2.75 | 14.72 ± 15.65 | 0.7877 |
| pythia-410m lr=0.5 | 34.69 ± 15.35 | 39.96 ± 1.86 | -5.27 ± 15.63 | 0.7883 |
| pythia-1.4b lr=0.5 | 37.41 ± 4.12 | 24.03 ± 2.49 | 13.37 ± 5.17 | 0.8315 |
| pythia-1.4b Ŗ lr=0.5 | 38 ± 4.72 | 24.74 ± 2.49 | 13.27 ± 5.89 | 0.8454 |
| SmolLM2-135M lr=0.1 | 60.79 ± 17.6 | 74.04 ± 11.45 | -13.25 ± 23.9 | 0.8475 |
| SmolLM2-360M lr=0.1 | 65.71 ± 16.07 | 69.05 ± 9.6 | -3.34 ± 19.42 | 0.9843 |
| SmolLM2-1.7B lr=0.1 | 49.85 ± 2.75 | 56.47 ± 9.19 | -6.62 ± 9.32 | 0.9532 |
| SmolLM2-1.7B Ŗ lr=0.1 | 49.95 ± 2.82 | 56.47 ± 9.19 | -6.52 ± 9.44 | 0.9535 |
| gemma-3-270m lr=0.1 | 12.94 ± 3.04 | 60.87 ± 1.43 | -47.93 ± 3.87 | 0.3448 |
| gemma-3-1b-pt lr=0.1 | 18.67 ± 3.14 | 61.27 ± 1.73 | -42.6 ± 3.01 | 0.0998 |
| gemma-3-4b-pt lr=0.1 | 7.97 ± 2.09 | 61.17 ± 1.02 | -53.19 ± 2.05 | 0.0497 |
| gemma-3-4b-pt Ŗ lr=0.1 | 6.84 ± 0.4 | 63.56 ± 1.02 | -56.72 ± 1.22 | -0.1461 |

*Table 27.* Progressive cramming with and without the BOS token. Ŗ denotes runs where the BOS token is removed.

| Model | Cram Tokens | Info Gain |
|---|---|---|
| Llama-3.1-8B lr=0.1 | 1064 ± 394 | 3028 ± 1321 |
| Llama-3.1-8B Ŗ lr=0.1 | 1412 ± 320 | 4473 ± 1035 |
| pythia-1.4b lr=0.5 | 544 ± 57 | 1694 ± 125 |
| pythia-1.4b Ŗ lr=0.5 | 541 ± 68 | 1687 ± 121 |
| SmolLM2-1.7B lr=0.1 | 370 ± 113 | 1119 ± 350 |
| SmolLM2-1.7B Ŗ lr=0.1 | 361 ± 105 | 1106 ± 328 |
| gemma-3-4b-pt lr=0.1 | 286 ± 165 | 949 ± 466 |
| gemma-3-4b-pt Ŗ lr=0.1 | 207 ± 86 | 1196 ± 445 |

### F.3. Model-Width Ablation

The depth ablation (Appendix F.1) varies the number of layers at a fixed width; here we hold depth fixed and vary model *width*. Across the SmolLM2 family (135M, 360M, and 1.7B; hidden sizes 576, 960, and 2048) we keep only the first 4 and last 4 decoder layers of each model (8 layers total), so every checkpoint has identical depth and differs essentially only in width. As in the depth ablation, discarding the middle block leaves the retained layers stitched across a gap they were never trained to bridge, so we *recover* each truncated checkpoint before measuring compression. We finetune all parameters with a standard next-token objective on fineweb-edu for 5k optimizer steps at sequence length 1024 with an effective batch of ≈256k tokens per step (AdamW, cosine-with-min-lr schedule, peak learning rate $10^{-3}$, 500 warmup steps, bf16, torch.compile; 8×A100), then run baseline progressive cramming with random per-sample initialization.

Every progressive-cramming run uses the same PG19 sample set, learning rate (0.1), and optimization budget as our main SmolLM2-1.7B run, so the only varying factor is model width. Table 31 reports, for each width, the achieved compressed-token count, information gain, trajectory length, and the number of principal components explaining 99% of trajectory variance (PCA 99%). Concretely, the mean number of perfectly reconstructed (compressed) tokens rises sharply with width: 42 at 135M, 84.5 at 360M, and 428.6 at 1.7B.

We then run the baseline progressive-cramming procedure on each checkpoint, using the same PG19 sample set, learning rate (0.1), and optimization budget as our main run; the fully pretrained model is the reference. Table 30 reports, for each variant, the achieved compressed-token count, information gain, trajectory length, and the number of principal components explaining 99% of trajectory variance (PCA 99%).

The fully pretrained model compresses 335 tokens on average. Re-initializing the LM head causes by far the largest drop, to only 33 tokens, whereas re-initializing the input embeddings or the transformer layers is far less damaging, leaving 173 and 148 tokens respectively. Progressive cramming is therefore most sensitive to the LM head: a pretrained output projection matters more than either pretrained embeddings or pretrained layers.

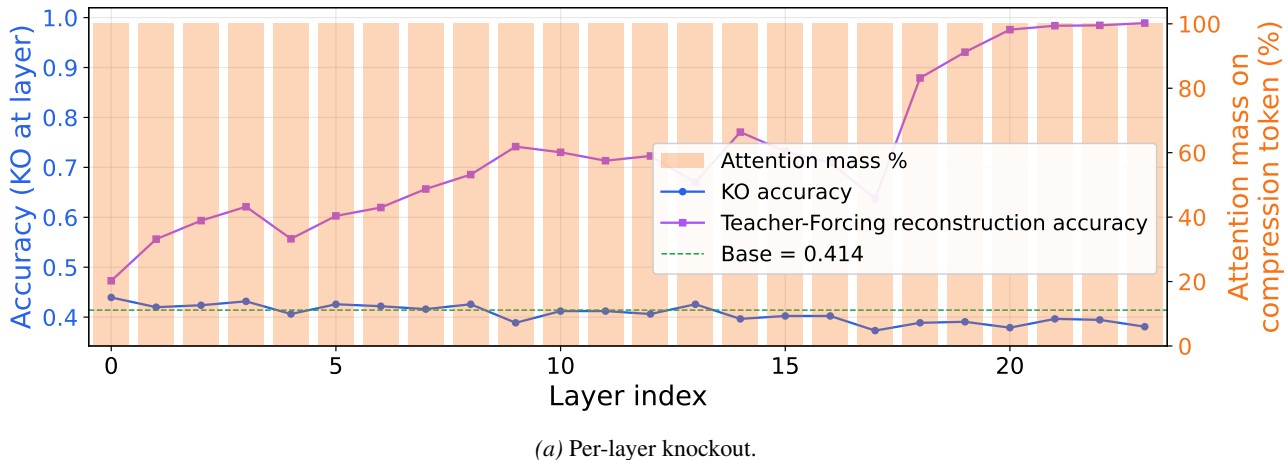

*(a)* Per-layer knockout.

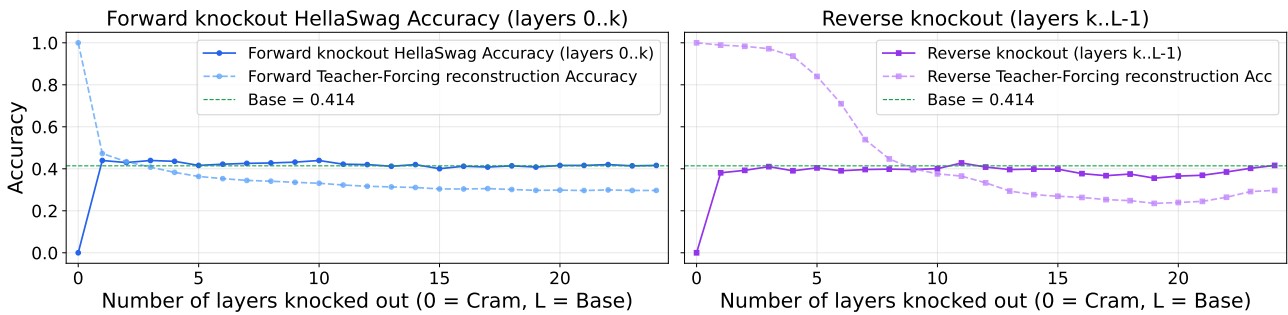

*(b)* Cumulative knockout (left: forward, masking layers 0 through $k$; right: reverse, masking layers $k$ through $L-1$).

*Figure 9.* Attention knockout on Pythia-1.4B ($L = 24$), using the protocols of Figures 6 and 7. The early-layer reconstruction collapse, the dissociation between attention mass and causal importance, and the forward/reverse asymmetry all reproduce the Llama-3.1-8B findings.

### F.4. Finetuning Sequence-Length Ablation

The causal-LM recovery route (Appendix F.3) finetunes each truncated checkpoint at sequence length 1024. Here we ask whether that finetuning sequence length affects the compression measured afterward. Starting from the same SmolLM2-1.7B first-4 + last-4 checkpoint, we re-finetune it with everything held fixed except the training sequence length—512, 1024 (the baseline), and 2048—and, at the off-baseline lengths, also vary the peak learning rate ($\frac{1}{2}\times$ and $1\times$ at 512; $1\times$ and $2\times$ at 2048) to check that any seq-length effect is not merely an LR artifact. To keep memory and the effective batch comparable, we hold the per-device token footprint and the 256-sequence global batch constant across variants (per-device batch $= 8192/\text{seq\_len}$). Every fine-tuned checkpoint is then evaluated with the identical base-line progressive-cramming procedure (random per-sample initialization, PG19, $\text{lr} = 0.1$), exactly as the causal-LM rows of Table 31.

Table 32 shows that the finetuning sequence length has no meaningful effect on compression. The mean number of perfectly reconstructed tokens is 414.1 and 404.5 at seq 512 ($\text{lr} \, 5 \times 10^{-4}$ / $10^{-3}$), 428.6 at the seq-1024 baseline, and

434.4 and 390.7 at seq 2048 ($\text{lr} \, 10^{-3}$ / $2 \times 10^{-3}$)—every variant lands within $\approx$38 tokens of the baseline, well inside the per-run standard deviation ($\approx$60–77). If anything the learning rate matters slightly more than the sequence length (the two seq-2048 rows, differing only in LR, bracket the baseline), but neither effect is statistically meaningful here. Compression capacity is thus insensitive to the finetuning sequence length over this $4\times$ range.

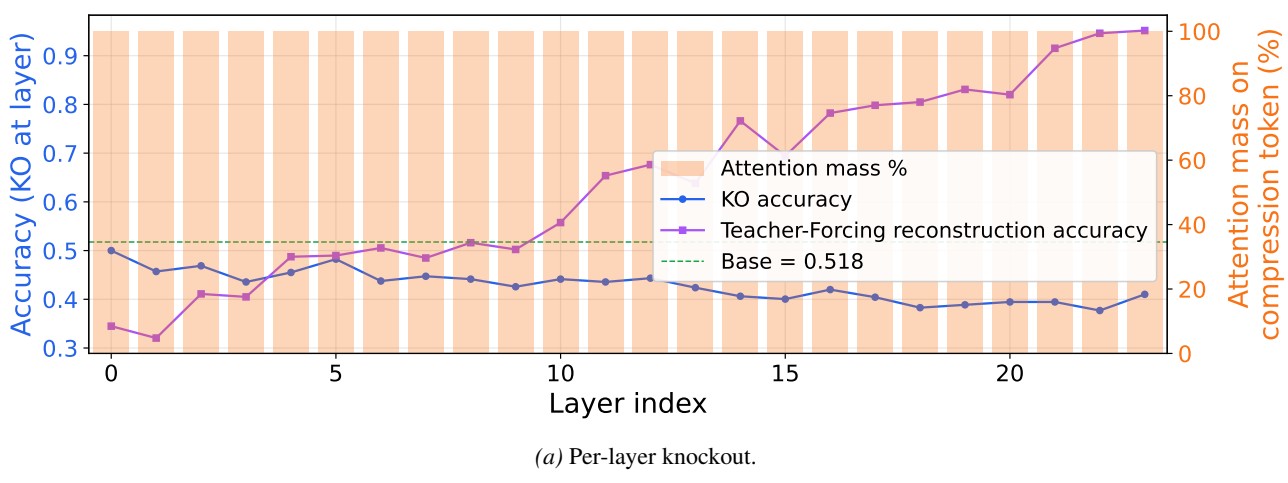

*(a)* Per-layer knockout.

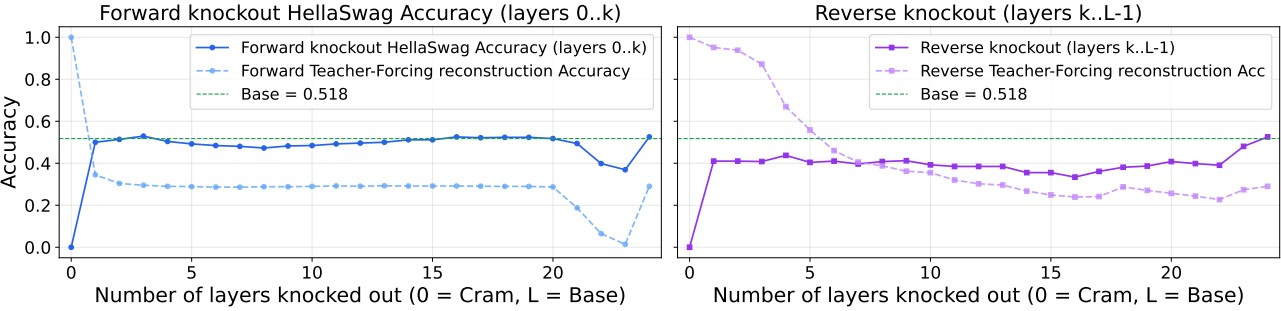

*(b)* Cumulative knockout (left: forward, masking layers $0$ through $k$; right: reverse, masking layers $k$ through $L-1$).

*Figure 10.* Attention knockout on SmolLM2-1.7B ($L = 24$), using the protocols of Figures 6 and 7. The findings are consistent with Llama-3.1-8B: early-layer interactions causally drive the downstream collapse, while the high-attention-mass late layers do not.

*Table 28.* Compression capacity vs. network depth and model size (full breakdown of the main-text pivot, Table 5). Each pretrained model (name annotated with its total decoder-layer count) is truncated to its first $N$ decoder layers, recovered by a short causal-LM finetune on fineweb-edu, and evaluated with progressive cramming (PG19, 50 samples, sequence length 4096, lr = 0.1, random per-sample initialization); "full" is the untruncated model. "Compressed Tokens" is the achieved prefix length $n$, "Trajectory Length" is $L_{\text{traj}}$, and "PCA 99%" is the number of principal components explaining 99% of trajectory variance. Compression rises monotonically with both retained depth (within each family) and model size (across families at matched depth).

| Model | Compressed Tokens | Information Gain | Trajectory Length | PCA 99% |
|---|---|---|---|---|
| SmolLM2-1.7B (24L) first 1 | $50 \pm 5.9$ | $301 \pm 26$ | $627 \pm 107$ | $17.12 \pm 1.56$ |
| SmolLM2-1.7B (24L) first 2 | $240.5 \pm 30$ | $1291 \pm 134$ | $1966 \pm 195$ | $31.3 \pm 1.78$ |
| SmolLM2-1.7B (24L) first 4 | $404.5 \pm 58.2$ | $2058 \pm 160$ | $2120 \pm 223$ | $31.82 \pm 2.05$ |
| SmolLM2-1.7B (24L) first 8 | $455 \pm 68.5$ | $2172 \pm 191$ | $1552 \pm 318$ | $31.46 \pm 3.38$ |
| SmolLM2-1.7B (24L) full | $335.1 \pm 61.3$ | $1208 \pm 162$ | $1051 \pm 147$ | $33.22 \pm 2.82$ |
| SmolLM3-3B (36L) first 1 | $20.2 \pm 4$ | $126 \pm 15$ | $253 \pm 60$ | $9.4 \pm 1.33$ |
| SmolLM3-3B (36L) first 2 | $38.7 \pm 7.3$ | $216 \pm 20$ | $485 \pm 75$ | $14.08 \pm 1.43$ |
| SmolLM3-3B (36L) first 4 | $113.8 \pm 13.2$ | $566 \pm 45$ | $960 \pm 141$ | $22.64 \pm 1.66$ |
| SmolLM3-3B (36L) first 8 | $300.1 \pm 40.6$ | $1390 \pm 127$ | $1928 \pm 209$ | $29.88 \pm 1.14$ |
| Qwen3-4B (36L) first 1 | $79.3 \pm 13.2$ | $460 \pm 47$ | $1078 \pm 101$ | $18.96 \pm 1.15$ |
| Qwen3-4B (36L) first 2 | $146.7 \pm 23.7$ | $792 \pm 87$ | $1672 \pm 148$ | $26.6 \pm 1.43$ |
| Qwen3-4B (36L) first 4 | $192.3 \pm 34.3$ | $991 \pm 128$ | $1564 \pm 182$ | $29.1 \pm 1.73$ |
| Qwen3-4B (36L) first 8 | $420.8 \pm 55.1$ | $2007 \pm 155$ | $2473 \pm 203$ | $37.32 \pm 1.41$ |
| Qwen3-4B (36L) full | $512 \pm 105.2$ | $2389 \pm 324$ | $2043 \pm 269$ | $47.28 \pm 4.69$ |
| Qwen3-8B (36L) first 1 | $119 \pm 15.9$ | $680 \pm 60$ | $1739 \pm 167$ | $22.72 \pm 1.06$ |
| Qwen3-8B (36L) first 2 | $179.9 \pm 25.9$ | $963 \pm 96$ | $2384 \pm 187$ | $28.92 \pm 1.09$ |
| Qwen3-8B (36L) first 4 | $303.5 \pm 46.6$ | $1552 \pm 153$ | $2864 \pm 216$ | $33.76 \pm 1.35$ |
| Qwen3-8B (36L) first 8 | $624.5 \pm 79.4$ | $2969 \pm 176$ | $4170 \pm 251$ | $42.98 \pm 1.39$ |
| Qwen3-8B (36L) full | $774.2 \pm 158.9$ | $3093 \pm 453$ | $3380 \pm 585$ | $53.9 \pm 5.94$ |
| Llama-3.1-8B (32L) first 1 | $96.7 \pm 10.7$ | $545 \pm 41$ | $1447 \pm 134$ | $31 \pm 1.52$ |
| Llama-3.1-8B (32L) first 2 | $183.8 \pm 31.5$ | $896 \pm 116$ | $2225 \pm 337$ | $33.12 \pm 2.57$ |
| Llama-3.1-8B (32L) first 4 | $399.7 \pm 104$ | $1858 \pm 404$ | $3031 \pm 367$ | $36.02 \pm 1.35$ |
| Llama-3.1-8B (32L) full | $1437.6 \pm 380.1$ | $4391 \pm 1408$ | $6174 \pm 1263$ | $82.78 \pm 9.07$ |

*Table 29.* Detailed transformer-depth cuts supporting the depth–size table (Table 28). *First+last*: SmolLM2-1.7B with the first $N$ and last $N$ decoder layers ($2N$ total), shown as-is and after causal-LM finetuning. *Last-only*: SmolLM2-1.7B and Llama-3.1-8B with only the last $N$ layers (finetuned). Full-depth references are in Table 28. "Compressed Tokens" is the achieved prefix length $n$, "Trajectory Length" is $L_{\text{traj}}$, and "PCA 99%" is the number of principal components explaining 99% of trajectory variance.

| Model | Compressed Tokens | Information Gain | Trajectory Length | PCA 99% |
|---|---|---|---|---|
| SmolLM2-1.7B first+last 2 layers | $76.9 \pm 39.3$ | $2082 \pm 1285$ | $1060 \pm 345$ | $23.12 \pm 5.14$ |
| SmolLM2-1.7B first+last 2 layers (finetuned) | $233.8 \pm 26$ | $1261 \pm 98$ | $1964 \pm 190$ | $35.62 \pm 1.47$ |
| SmolLM2-1.7B first+last 4 layers | $270.9 \pm 50.4$ | $5198 \pm 857$ | $1665 \pm 253$ | $32.48 \pm 1.99$ |
| SmolLM2-1.7B first+last 4 layers (finetuned) | $453.3 \pm 60.5$ | $2344 \pm 125$ | $2255 \pm 182$ | $37.18 \pm 1.81$ |
| SmolLM2-1.7B first+last 8 layers | $240.8 \pm 54.7$ | $5585 \pm 1998$ | $1036 \pm 234$ | $27.78 \pm 3.37$ |
| SmolLM2-1.7B first+last 8 layers (finetuned) | $428.6 \pm 67.2$ | $2029 \pm 194$ | $1414 \pm 209$ | $30.92 \pm 2.89$ |
| SmolLM2-1.7B first+last 16 layers | $272.5 \pm 38.3$ | $1544 \pm 223$ | $1083 \pm 120$ | $34.48 \pm 3.09$ |
| SmolLM2-1.7B first+last 16 layers (finetuned) | $335.3 \pm 60.2$ | $1407 \pm 198$ | $1030 \pm 164$ | $29.42 \pm 3.23$ |
| SmolLM2-1.7B last-only 1 layer | $88.8 \pm 10.9$ | $551 \pm 51$ | $1412 \pm 167$ | $24.02 \pm 1.81$ |
| SmolLM2-1.7B last-only 2 layers | $176 \pm 22.9$ | $1025 \pm 103$ | $1801 \pm 224$ | $28.54 \pm 2.27$ |
| SmolLM2-1.7B last-only 4 layers | $211.7 \pm 33.6$ | $1202 \pm 130$ | $1556 \pm 178$ | $29.42 \pm 2.38$ |
| SmolLM2-1.7B last-only 8 layers | $351.5 \pm 42.8$ | $1798 \pm 145$ | $1437 \pm 124$ | $28.16 \pm 1.65$ |
| Llama-3.1-8B last-only 1 layer | $56.3 \pm 8.3$ | $347 \pm 40$ | $1353 \pm 171$ | $18.64 \pm 1.68$ |
| Llama-3.1-8B last-only 2 layers | $116.9 \pm 18.8$ | $630 \pm 62$ | $2305 \pm 242$ | $27.94 \pm 1.87$ |
| Llama-3.1-8B last-only 4 layers | $330.3 \pm 44.6$ | $1661 \pm 153$ | $3384 \pm 253$ | $38.02 \pm 1.76$ |

*Table 30.* Initialization ablation on SmolLM2-1.7B. Exactly one component is randomly re-initialized—transformer layers, LM head, or input embeddings—while the rest stay pretrained; the fully pretrained model is the reference. For the LM-head and input-embedding variants the tied embeddings are first untied so a single matrix is randomized. "Compressed Tokens" is the achieved prefix length $n$, "Trajectory Length" is $L_{\text{traj}}$, and "PCA 99%" is the number of principal components explaining 99% of trajectory variance.

| Model | Compressed Tokens | Information Gain | Trajectory Length | PCA 99% |
|---|---|---|---|---|
| Pretrained | $335.1 \pm 61.3$ | $1208 \pm 162$ | $1051 \pm 147$ | $33.22 \pm 2.82$ |
| Random transformer layers | $148.5 \pm 11.8$ | $4396 \pm 579$ | $1655 \pm 105$ | $35.6 \pm 2$ |
| Random LM head | $33.6 \pm 7.4$ | $503 \pm 108$ | $216 \pm 37$ | $13.52 \pm 1.64$ |
| Random input embeddings | $173.9 \pm 35.2$ | $2532 \pm 508$ | $832 \pm 100$ | $35.1 \pm 2.78$ |

*Table 31.* Model-width ablation on the SmolLM2 family. Every model is truncated to its first 4 and last 4 decoder layers (8 layers total), holding depth fixed while width varies (hidden sizes 576, 960, 2048). Each width is recovered by plain causal-LM finetuning before measuring compression (evaluated with baseline random per-sample initialization). "Compressed Tokens" is the achieved prefix length $n$, "Trajectory Length" is $L_{\text{traj}}$, and "PCA 99%" is the number of principal components explaining 99% of trajectory variance.

| Model | Compressed Tokens | Information Gain | Trajectory Length | PCA 99% |
|---|---|---|---|---|
| 135M (causal-LM) | $42 \pm 15.9$ | $253 \pm 84$ | $222 \pm 64$ | $14.34 \pm 3.49$ |
| 360M (causal-LM) | $84.5 \pm 20$ | $465 \pm 100$ | $335 \pm 61$ | $18.6 \pm 2.5$ |
| 1.7B (causal-LM) | $428.6 \pm 67.2$ | $2029 \pm 194$ | $1414 \pm 209$ | $30.92 \pm 2.89$ |

*Table 32.* Finetuning sequence-length ablation on SmolLM2-1.7B (first-4 + last-4 = 8 layers). The same truncated checkpoint is re-finetuned on fineweb-edu at sequence lengths 512 / 1024 / 2048 (and varying learning rate), holding the per-device token footprint and 256-sequence global batch constant, then evaluated with baseline progressive cramming (PG19, lr = 0.1). The seq-1024 / lr $10^{-3}$ row is the Table 31 causal-LM baseline. "Compressed Tokens" is the achieved prefix length $n$, "Trajectory Length" is $L_{\text{traj}}$, and "PCA 99%" is the number of principal components explaining 99% of trajectory variance.

| Model | Compressed Tokens | Information Gain | Trajectory Length | PCA 99% |
|---|---|---|---|---|
| seq 512 / lr 0.0005 | $414.1 \pm 60.1$ | $2048 \pm 152$ | $1375 \pm 208$ | $31.66 \pm 3.1$ |
| seq 512 / lr 0.001 | $404.5 \pm 69.7$ | $1936 \pm 200$ | $1332 \pm 222$ | $30.22 \pm 3.02$ |
| seq 1024 / lr 0.001 (baseline) | $428.6 \pm 67.2$ | $2029 \pm 194$ | $1414 \pm 209$ | $30.92 \pm 2.89$ |
| seq 2048 / lr 0.001 | $434.4 \pm 61.6$ | $2049 \pm 235$ | $1498 \pm 231$ | $32.38 \pm 2.51$ |
| seq 2048 / lr 0.002 | $390.7 \pm 77.3$ | $1823 \pm 211$ | $1257 \pm 225$ | $29.4 \pm 2.49$ |

