# OpenReview forum: "Progressive Cramming: Reliable Token Compression and What It Reveals"
_ICML.cc/2026/Conference — ICML 2026 regular_

### Official Review · Reviewer_sozN · 2026-02-28

**Soundness:** 3
**Presentation:** 3
**Significance:** 3
**Originality:** 3
**Overall Recommendation:** 5
**Confidence:** 4

**Summary:**

The work addresses an optimization approach to compression of token sequences. A new method of progressive optimization for perfect sequence reconstruction is proposed. Along with this, activation alignment and low-dimensional projection are proposed and show effectiveness. Authors address weaknesses of the previous work, namely, not 100% reconstruction accuracy. Furthermore, authors show that the optimization trajectory occupies a relatively low-dimensional manifold. The paper clearly investigates and explains the underlying mechanics of token compression via attention sinks and attention hijacking. Authors show that the representations obtained by optimization of next token prediction cannot be applied to downstream tasks in a straightforward manner.

**Compliance With Llm Reviewing Policy:**

Affirmed.

**Final Justification:**

The final score is 5 (Accept). Authors propose a novel method for optimization-based lossless sequence embedding method, clearly show the limitations of the method on downstream tasks, and discover the underlying mechanism of the failure modes. Authors fully addressed the concerns raised in the review, therefore, the overall recommendation is kept positive.

**Key Questions For Authors:**

1. How does the speed (in terms of optimization steps) of progressive optimization relates to the vanilla cramming?
2. How semantic properties of the resulting representations can be evaluated other than by concatenation with the queries?
3. How can the number of optimization steps be reduced?

**Limitations:**

yes

**Strengths And Weaknesses:**

Strengths:
- a novel method of cramming based on the progressive optimization is proposed
- in depth analysis of the underlying nature of the token compression based on the analysis of optimization trajectories and attention patterns
- two novel approaches for regularization of token compression are proposed
- evaluation of practical applicability of a novel cramming method
- clear and better motivated evaluation protocol compared to prior work
Weaknesses:
- single dataset used for evaluation, some extra datasets or ones with more diverse distribution of the language might be beneficial for the work
- no proposal for introducing valuable semantic properties, but statement of the non-applicability
- higher cost of progressive optimization procedure

---

> ### Author Rebuttal · Authors · 2026-03-30
>
> We thank the reviewer for the thoughtful comments and suggestions for strengthening the paper. Our responses follow below.
>
>
>
> # Weaknesses:
>
> **W1. Single evaluation dataset.**
>
> To address this issue we reproduced fanfics dataset from Kuratov et al and runned experiments on it. Results may be found in [Table 11](https://drive.google.com/file/d/1iggPKMXXLJ_uL018BLOI3Ummv780ayF7/view?usp=sharing).
>
> The key findings are consistent with those observed on PG-19. Information gain follows the same patterns across configurations.
>
> The low-dimensional structure of compression trajectories also generalizes to Fanfics data. While PCA 99% values are somewhat larger than those observed on PG-19-likely reflecting the greater distributional diversity of fan-fiction text-they remain at least an order of magnitude below the embedding dimensionality.
>
>
>
>
> **W2. No proposal for introducing valuable semantic properties**
>
> We agree-cramming optimizes solely for token reconstruction and does not explicitly
> target semantic properties. Introducing such properties is outside the scope of this
> work. Compression approaches that do optimize for semantics (e.g., Gisting, Activation
> Beacon, AutoCompressors, ICAE) are discussed in the related work section and represent
> a complementary research direction.
>
>
>
>
> **W3. Higher cost of progressive optimization procedure**
>
> Progressive cramming optimization is about $2\times$ slower than full cramming. The main
> bottleneck in current implementation is that within a batch, a single slowly converging sample blocks new token addition for all other samples.
>
>
>
>
> ---
>
>
>
>
> # Questions:
>
> **Q1. How does the speed (in terms of optimization steps) of progressive optimization relate to vanilla cramming?**
>
> Full cramming was limited to 10k steps. Progressive cramming, due to its autoregressive approach, may finish earlier. On average, optimization steps remain below 10k for progressive cramming:
>
> * Pythia-1.4B:   $6844 \pm 2043$
> * SmolLM2-1.7B:  $9391 \pm 2068$
> * Gemma-3-4B-pt: $8954 \pm 2900$
>
> Experiments for Llama 8B were not conducted due to time constraints.
>
> **Q2. How semantic properties of the resulting representations can be evaluated other than by concatenation with the queries?**
>
> **Generative benchmark.** We additionally report evaluation results on **5-shot MMLU** in [Table 9](https://drive.google.com/file/d/1_NJSBkvhEyUdA45HD2QD8IxIEaQEpb9t/view?usp=sharing). Compressing only few-shot examples causes modest degradation ($-2\%$ to $-15\%$), while compressing the full prefix leads to complete failure ($0\%$ valid answers), consistent with our findings on likelihood-based benchmarks. See also answer to reviewer $\texttt{eppq}$, W3.
>
> **Different perplexity computation strategies.** We evaluated the semantic properties of compression embeddings on the **HellaSwag** and **ARC** benchmarks using different perplexity computation strategies. Specifically, we varied which token logits were included when computing perplexity:
>
>
> * **baseline** – standard perplexity, no compression embeddings
> * **compression** – perplexity computed with compression embeddings
> * **compression only** – perplexity computed using only compression embeddings in place of the context
>
> There are also **\*_endings** experiments modifications where perplexity was computed only with answer logits.
>
> By comparing performance across these methods, we can isolate the contributions of context versus compression tokens and evaluate the robustness of semantic representations in compressed form (see [Table 10](https://drive.google.com/file/d/1-qex3vN5c7WCSDDHi2bbBT29Cgy0oYZO/view?usp=sharing)).
>
> Based on this results we can conclude that **compression** is consistently better on all benchmarks than **compression only**. While **\*_endings** experiments are inconsistent across diffetent experiments setting and benchmarks.
>
> **Q3. How can the number of optimization steps be reduced?**
>
> Our low-dimensional projection partially addresses this-it converges faster and compresses fewer tokens. Further reduction could involve training an encoder to predict a good starting point, left for future work.
>
>
>
>
> ---
>
>
>
>
> **Erratum: Downstream evaluation bug**
>
> After submission, we discovered a bug in the perplexity computation for the HellaSwag and ARC benchmarks. The compression embedding introduced an off-by-one token shift that was not accounted for during likelihood scoring, causing the near-random downstream accuracy reported in the original paper. After correction, HellaSwag accuracy under cramming ranges from 34%-38% for most models, representing a consistent but moderate drop from baseline in line with our central claim that high reconstruction accuracy does not imply preservation of downstream-relevant semantics. Corrected results are presented in [Table 7](https://drive.google.com/file/d/1DIjiTpyIqQ4xuW0k9Q7mjCOFuXUPz2YX/view?usp=sharing). See also response to Reviewer $\texttt{neRn}$, Q3.

---

> > ### Author Rebuttal · Reviewer_sozN · 2026-04-04
> >
> > Thank you for the detailed response. The concerns raised in the review are fully resolved, therefore, I keep the score positive.

---

### Official Review · Reviewer_UoVE · 2026-03-06

**Soundness:** 2
**Presentation:** 2
**Significance:** 3
**Originality:** 3
**Overall Recommendation:** 3
**Confidence:** 5

**Summary:**

This paper introduces progressive cramming as an alternative to full fixed-budget cramming. Prior work by Kuratov et al. shows that a single learned vector can compress up to 1568 tokens in Llama-3.1-8B, but reconstruction accuracy typically peaks at around 99%, and small early-token errors can cause autoregressive generation to collapse. The central contribution is a token-by-token optimization procedure that enforces exact (100%) reconstruction by trading off compressed length and information gain for correctness. The method is characterized across multiple model families through trajectory analysis, attention measurements, and several ablations.

**Compliance With Llm Reviewing Policy:**

Affirmed.

**Final Justification:**

I honestly really like this paper, and I would have given it a high score, if not for the distracting and unconvincing attention-hijacking narrative (which takes up a large part of the paper). If the manuscript had stayed more tightly focused on progressive cramming itself, I think this would have been an immediate accept for me. My main issue is that too much of the paper is spent on this mechanistic story that does not yet feel sufficiently established, and the downstream evaluation bug further reinforces the need for a more careful and coherent presentation. For that reason, I am maintaining my current score, while encouraging the authors to revise and resubmit.

**Key Questions For Authors:**

I understood the content sufficiently.

**Limitations:**

Yes.

**Strengths And Weaknesses:**

**Soundness:** The progressive cramming protocol is technically sound and addresses a real weakness in prior cramming evaluations by defining compression limits through an exact 100% reconstruction criterion (Sec. 4.1). The empirical study spans multiple model families and includes meaningful ablations of optimization and regularization choices (Tables 1-2; Sec. 5.1-5.4). However, the attention-hijacking narrative, which takes up roughly 15 to 25% of the main text, weakens the paper (see core weakness below). While the protocol itself is well supported, the mechanistic claims surrounding attention concentration are less clearly established.


**Presentation:** The core idea of progressive cramming is presented clearly. However, the paper drifts into experimental threads that are not tightly integrated with the central contribution, particularly the extended attention-hijacking analysis (see core weakness below). The manuscript would benefit from a tighter narrative, clearer consolidation of results, and more explicit prioritization of the progressive protocol as the main takeaway. In its current form, the paper feels somewhat diffuse, and certain sections could have been developed or reorganized to better support the central claim. These issues are addressable in a revised version, but they limit the clarity and overall polish of the submission.


**Originality:** The paper builds directly on prior token cramming work but makes a meaningful step forward by demonstrating exact (100%) reconstruction rather than stopping at ~99%. While the general setting is not new, this refinement is a substantive methodological improvement. However, portions of the manuscript move away from this central contribution (see core weakness below), which makes the overall originality feel less focused than it could be.


**Significance:** The paper demonstrates exact reconstruction for compression in frozen LLMs and shows that a progressive curriculum enables a single learned vector to achieve lossless recovery. This advances our understanding of representational capacity and provides a useful protocol for studying compression limits. However, the extensive focus on the attention-hijacking narrative diffuses the impact of this result. By shifting emphasis away from the compression protocol itself, the paper partially obscures its strongest contribution.

**_Core weakness:_**   A major weakness is the attention-hijacking narrative (Sec. 4.4 and Sec. 5.5; Tables 3-4), which occupies a substantial portion of the main paper. As written, this part of the manuscript reads more like an exploratory collection of observations than a cohesive mechanistic account, and it detracts from the paper’s stronger ideas about progressive compression. The authors emphasize strong model-family dependence, yet the aggregate attention-mass statistics in Table 3 do not yield a mechanism that clearly generalizes across families. The BOS ablation in Table 4 is a natural interference test, but its effects are inconsistent across models, with large gains for Llama, negligible changes for Pythia and SmolLM2, and mixed behavior for Gemma. This makes it difficult to sustain a single “competing attention sinks” explanation. The downstream degradation results (Sec. 5.6; Table 5) are presented alongside this narrative, yet the link between attention concentration and capability collapse is not clearly established.

---

> ### Author Rebuttal · Authors · 2026-03-30
>
> We are grateful for the detailed and insightful review. We address the raised concerns.
>
>
>
>
> # Weaknesses:
>
> **W1. Attention-hijacking narrative**
>
> We conducted attention knockout experiments (see answer to reviewer $\texttt{neRn}$, W2):
> the forward/reverse knockout asymmetry shows that early-layer interactions with the compression embedding drive downstream degradation, while late-layer attention mass has minimal causal impact. And observed attention concentration possibly is a symptom rather than the cause.
>
>
>
>
>
> ---
>
>
>
>
> **Erratum: Downstream evaluation bug**
>
> After submission, we discovered a bug in the perplexity computation for the HellaSwag and ARC benchmarks. The compression embedding introduced an off-by-one token shift that was not accounted for during likelihood scoring, causing the near-random downstream accuracy reported in the original paper. After correction, HellaSwag accuracy under cramming ranges from 34%-38% for most models, representing a consistent but moderate drop from baseline in line with our central claim that high reconstruction accuracy does not imply preservation of downstream-relevant semantics. Corrected results are presented in [Table 7](https://drive.google.com/file/d/1DIjiTpyIqQ4xuW0k9Q7mjCOFuXUPz2YX/view?usp=sharing). See also response to Reviewer $\texttt{neRn}$, Q3.

---

> > ### Author Rebuttal · Reviewer_UoVE · 2026-03-31
> >
> > This paper has clear potential. However, I do not support acceptance in its current form, as the manuscript would benefit from substantial reorganization. In particular, the authors’ note about the downstream evaluation bug further highlights the value of stepping back and presenting the results within a more coherent and focused narrative. I encourage the authors to revise and resubmit this work in a future cycle after reorganizing the paper accordingly.

---

> > > ### Author Response · Authors · 2026-04-03
> > >
> > > We thank the reviewer for the valuable feedback and the opportunity to clarify.
> > >
> > >
> > >
> > >
> > >
> > >
> > > **On the downstream evaluation bug.**
> > >
> > > The bug affected absolute accuracy values (corrected HellaSwag: 34-38% instead of ~25%), but the core finding holds: cramming consistently degrades downstream performance despite high reconstruction accuracy. We also evaluated on 5-shot MMLU (see response to Reviewer $\texttt{eppq}$, W3), where full-prefix compression causes complete generation failure. This confirms that the degradation is real and not an artifact of the bug or evaluation protocol.
> > >
> > >
> > >
> > >
> > >
> > > **On paper reorganization.**
> > >
> > > We agree that the presentation can be improved and will reorganize the manuscript in the revised version. We believe that the core contributions - the identification of the "last 1%" problem, progressive cramming, and the systematic evaluation of downstream performance under compression - provide sufficient evidence to support the paper's conclusions.

---

### Official Review · Reviewer_eppq · 2026-03-09

**Soundness:** 3
**Presentation:** 2
**Significance:** 2
**Originality:** 3
**Overall Recommendation:** 3
**Confidence:** 3

**Summary:**

The paper proposes progressive cramming, which incrementally extends the reconstruction target token by token and requires 100% reconstruction, replacing earlier protocols based on fixed token budgets and 99% teacher-forcing accuracy. It shows that cramming trajectories exhibit strong low-dimensional structure, while the learned compression embedding often becomes an attention sink in specific intermediate layers. The authors conduct extensive experiments across diverse datasets and models demonstrating the good performance of this proposed method.

**Compliance With Llm Reviewing Policy:**

Affirmed.

**Final Justification:**

I would maintain my weak reject score as this paper would benefit from substantial reorganization like fix the evaluation bug and discuss more about the efficiency.

**Key Questions For Authors:**

Please refer to the weakness above.

**Limitations:**

yes

**Strengths And Weaknesses:**

Strength
1. The paper is well written and easy to follow and the authors conducted comprehensive experiments.
2. The paper identifies that 99% teacher-forcing accuracy does not imply usable autoregressive reconstruction, especially when early-token errors trigger cascading failures. It gives motivation for progressive cramming.
3. The paper provides comprehensive analysis that further explains what cramming is actually doing.

Weakness
1. The experimental sample size is quite small that most experiments use only 10 PG19 samples. For high-variance phenomena such as trajectory PCA statistics, this substantially limits statistical reliability. Moreover, many conclusions are presented as mean ± std, but with such a small sample size it remains unclear whether the averages are stable or dominated by a few difficult instances.
2. The paper uses full cramming rather than progressive cramming on HellaSwag and ARC, and explicitly states that imperfect cases retained in the results. It is difficult to disentangle whether the capability drop is caused by the crammed embedding itself disrupting reasoning or simply by failed reconstruction corrupting the prefix.
3. The downstream evaluation remains relatively narrow, relying only on HellaSwag and ARC. While the authors argue that short-context tasks better isolate capability failures, this setup still only probes a limited slice of downstream behavior.
4. The appendix reports roughly 2800 GPU-hours, while the main conclusions are still based on a small number of samples. I concern that the trade-off between compute expenditure and statistical coverage seems suboptimal. Broader validation on more samples might have been more convincing than extremely heavy optimization on a tiny set.
5. The paper does not fill eight pages, and although there is some additional content in the appendix, I recommend that the author reorganize the results to make the main body more substantial.

---

> ### Author Rebuttal · Authors · 2026-03-30
>
> We appreciate the reviewer's careful reading and valuable feedback. Below we respond to each concern.
>
>
>
>
> # Weaknesses:
>
> **W1. Limited number of samples.**
>
> We agree that 10 samples are statistically limited, so we re-ran all experiments on the
> 50-sample subset from Kuratov et al. Updated
> results are in [Table 6](https://drive.google.com/file/d/148Cxg2jOx_-MIH5XC0I-oPbwVLG0Ktiz/view?usp=sharing).
>
> Also see **W1** in our answer to reviewer $\texttt{neRn}$.
>
>
>
>
>
>
> **W2. Imperfect compression for HellaSwag and ARC evaluation**
>
> Due to HellaSwag and ARC benchmarks has short sequences (mostly less than 100 tokens), it would not be a problem to cram the sequences. Below we report the portion of fully converged samples for SmolLM2 (full cramming):
> * HellaSwag: 96%
> * ARC-Easy: 98%
> * ARC-Challenge: 97%
>
> In [Table 8](https://drive.google.com/file/d/1B4ALS1umyqq70ScHf-_1XtVp04rdG6eY/view?usp=sharing) we report benchmarks results only for fully converged samples for SmolLM2 model (full cramming setup). Imperfect reconstruction has minimal impact on benchmark performance. This confirms that the observed degradation is driven primarily by the compression embedding itself, rather than by reconstruction errors.
>
> We also evaluated percent of fully converged samples and benchmarks results for progressive cramming in [Table 7](https://drive.google.com/file/d/1DIjiTpyIqQ4xuW0k9Q7mjCOFuXUPz2YX/view?usp=sharing). For all models except gemma there is convergence above 95%. For more details see our response to reviewer $\texttt{neRn}$ to **Q1**.
>
>
>
>
>
> **W3. Narrow downstream evaluation relying only on HellaSwag and ARC**
>
> We agree that HellaSwag and ARC alone probe a limited slice of downstream behavior.
> To address this, we additionally evaluated on **5-shot MMLU** (512 samples) under
> three compression modes:
> * $\texttt{few-shot}$ - only few-shot examples are compressed
> * $\texttt{full-prefix}$ - both few-shot examples and the question are compressed
> * $\texttt{random}$ - a random embedding replaces the compression embedding as a control
>
> The results in [Table 9](https://drive.google.com/file/d/1_NJSBkvhEyUdA45HD2QD8IxIEaQEpb9t/view?usp=sharing) are consistent with our earlier findings. When only $\texttt{few-shot}$ tokens are
> compressed, all models shows accuracy degradation (ranging from
> $-2\%$ for Gemma-3-4B to $-15\%$ for Llama-3.1-8B), confirming that the model retains
> generative capability when uncompressed tokens follow the compression embedding. In
> contrast, $\texttt{full-prefix}$ compression where compression embedding compresses full prefix leads to **complete failure**: accuracy drops to
> near zero and the models produce no valid (parseable) answers, indicating total collapse
> of the generative process. Notably, the $\texttt{random}$ baseline causes negligible
> degradation, showing that the mere presence of an out-of-distribution
> token is not sufficient to explain the failure it is specifically the optimized
> compression embedding that disrupts downstream computation.
>
> These results on a generative benchmark with longer context reinforce the conclusions
> drawn from HellaSwag and ARC, and demonstrate that the downstream collapse generalizes
> beyond multiple-choice likelihood-based evaluation.
>
>
>
>
>
> **W4. Compute expenditure vs. statistical coverage**
>
> The reported 2800 GPU-hours represent total compute including all debugging,
> hyperparameter sweeps, and failed runs not the cost of final experiments alone.
>
> Progressive cramming is inherently expensive: current implementation was optimized through dynamic batching and code optimization, but it remains roughly $2\times$ slower than full cramming.
>
> As noted in our response to reviewer $\texttt{neRn}$ to **W1**, we have now extended all experiments to 50 samples following Kuratov et al., and the main conclusions remain stable [Table 6](https://drive.google.com/file/d/148Cxg2jOx_-MIH5XC0I-oPbwVLG0Ktiz/view?usp=sharing).
>
>
>
>
>
> **W5. Paper length and organization**
>
> We agree that the main body can be made more substantial. In the revised version we will
> reorganize the paper, ensuring the paper makes full use of the available page budget.
>
>
>
>
>
>
> **Erratum: Downstream evaluation bug**
>
> After submission, we discovered a bug in the perplexity computation for the HellaSwag and ARC benchmarks. The compression embedding introduced an off-by-one token shift that was not accounted for during likelihood scoring, causing the near-random downstream accuracy reported in the original paper. After correction, HellaSwag accuracy under cramming ranges from 34%-38% for most models, representing a consistent but moderate drop from baseline in line with our central claim that high reconstruction accuracy does not imply preservation of downstream-relevant semantics. Corrected results are presented in [Table 7](https://drive.google.com/file/d/1DIjiTpyIqQ4xuW0k9Q7mjCOFuXUPz2YX/view?usp=sharing). See also response to Reviewer $\texttt{neRn}$, Q3.

---

> > ### Author Rebuttal · Reviewer_eppq · 2026-04-02
> >
> > I sincerely thank the reviewer for the detailed response, it helps a lot, but I still have some remaining concern. Though 50 samples are more than 19, it is still a small number. And as the authors said, progressive cramming is inherently expensive, I think it needs more justification and explanation to convince me that it's worth it.

---

> > > ### Author Response · Authors · 2026-04-03
> > >
> > > We appreciate the reviewer's continued engagement. We address the remaining concerns below.
> > >
> > >
> > >
> > >
> > >
> > > **Why progressive cramming is necessary despite higher compute cost.**
> > >
> > > As discussed in Section 3.2, **full cramming** suffers from the *"last 1% problem"*: even at 97–99% teacher-forcing accuracy, the remaining errors concentrate at the earliest token positions. A single early-token mismatch causes autoregressive collapse - greedy convergence drops below 2%. Only **progressive cramming** reliably achieves 100% reconstruction accuracy from the compression embedding, which is a necessary condition for usable autoregressive generation.
> > >
> > >
> > >
> > >
> > >
> > > **More accurate estimation of embedding capacity.**
> > >
> > > Progressive cramming achieves an Information Gain of $4391 \pm 1408$, compared to $3292 \pm 320$ for full cramming (Kuratov et al.). The higher mean indicates that full cramming underestimates the true embedding capacity. The wider standard deviation is expected: progressive cramming pushes closer to the actual capacity limit, where sample-to-sample variation naturally increases.
> > >
> > >
> > >
> > >
> > >
> > > **Insights into the optimization landscape.**
> > >
> > > The progressive cramming setup allows us to trace the optimization trajectory, which revealed that it is inherently low-dimensional. This insight led to the low-rank projection technique that improves the cramming process, as demonstrated in [Table 6](https://drive.google.com/file/d/148Cxg2jOx_-MIH5XC0I-oPbwVLG0Ktiz/view?usp=sharing) (see our response to W1).
> > >
> > >
> > >
> > >
> > >
> > > **50 samples**
> > >
> > > We note that Kuratov et al. (full cramming) also evaluate on 50 samples. Our experimental setup therefore follows established practice in this line of work.

---

### Official Review · Reviewer_neRn · 2026-03-12

**Soundness:** 3
**Presentation:** 2
**Significance:** 4
**Originality:** 3
**Overall Recommendation:** 4
**Confidence:** 3

**Summary:**

This paper studies the Token Cramming phenomenon in large language models. The authors argue that prior evaluations based on a fixed token budget and 99% teacher-forcing accuracy do not adequately reflect true reconstruction ability, since even a few early-token errors can cause autoregressive generation to collapse. To address this issue, the paper proposes Progressive Cramming, which aims to characterize the reconstruction boundary more precisely and analyze the corresponding optimization trajectory. The results suggest that high reconstruction accuracy does not necessarily imply effective semantic compression, but may instead be associated with abnormal attention patterns induced by the compressed vector, which also substantially degrades downstream task performance.

**Compliance With Llm Reviewing Policy:**

Affirmed.

**Key Questions For Authors:**

1. The paper is based on a relatively small number of samples. Have the authors tested whether the main findings still hold on larger and more diverse datasets?
2. For attention hijacking, the current evidence appears mainly correlational. Could the authors include more direct intervention experiments to test whether attention concentration in specific layers or heads is actually important for reconstruction?
3. The paper argues that perfect reconstruction does not necessarily imply meaningful compression. Does this conclusion still hold when downstream evaluation is performed only on embeddings produced by Progressive Cramming?

**Limitations:**

yes

**Strengths And Weaknesses:**

Strengths
1. The paper has clear corrective value for the community. It usefully cautions that high reconstruction accuracy should not be equated with meaningful semantic compression, and that some reported token-cramming behavior may instead reflect a brittle attention-redistribution or steering effect.
2. The proposed Progressive Cramming protocol is a thoughtful analysis tool. Combined with PCA trajectory analysis, attention-mass tracking, and downstream evaluation, it forms a fairly coherent diagnostic pipeline for studying the phenomenon.

Weaknesses
1. As acknowledged by the authors, the empirical study is conducted on only a small number of PG19 samples, with some analyses based on very limited examples. For a paper that aims to reassess the community’s understanding of token cramming, this raises concerns about robustness and generalizability.
2. The evidence for “attention hijacking” remains primarily correlational: increased attention mass is observed alongside degraded downstream performance, but the paper lacks stronger causal interventions. For example, head/layer ablation, activation patching, or other targeted manipulations would help test whether disrupting this attention concentration actually impairs reconstruction or restores downstream capability.

---

> ### Author Rebuttal · Authors · 2026-03-30
>
> We thank the reviewer for the thorough evaluation and constructive suggestions.
> We address each point below.
>
>
>
>
> # Weaknesses
>
> **W1. Limited number of samples.**
>
> We reproduced the data preparation pipeline from Kuratov et al. and re-ran all experiments on the same 50 samples used in the original full cramming paper. We present the updated version of progressive cramming table [Table 6](https://drive.google.com/file/d/148Cxg2jOx_-MIH5XC0I-oPbwVLG0Ktiz/view?usp=sharing). Our main conclusions remain valid on this data.
>
> However, the expanded evaluation revealed an important finding: for all models
> except Llama-3.1-8B, activation alignment leads to significant degradation in
> both compressed tokens and Information Gain. The best results across all experiments
> are achieved with **low-dimensional linear projection alone**. Only Llama-3.1-8B
> shows competitive performance when combining low-dimensional projection with activation
> alignment; for the remaining models (Pythia-1.4B, SmolLM2-1.7B, Gemma-3-4B-pt) the
> combination does not improve over projection-only variants.
>
>
>
>
> **W2. Causality analysis for 'attention hijacking' and 'degraded downstream performance'**
>
> We thank the reviewer for this suggestion. We conducted causal intervention experiments
> using **attention knockout** - masking pre-softmax logits corresponding to the
> compression embedding to $-\infty$ at selected layers, thereby completely removing its
> influence on the residual stream. Three protocols were used: (1) **per-layer knockout**
> (masking at a single layer), (2) **forward cumulative knockout** (masking layers
> $ 0 $ through $ k $), and (3) **reverse cumulative knockout** (masking layers $ L{-}1 $
> down to $ k $). Each condition is evaluated on reconstruction accuracy and downstream
> capability (HellaSwag). Llama3.1-8B model was used for this ablation.
>
> Figures links:
> * [Per-layer knockout](https://drive.google.com/file/d/1nsBwmqWAdzuaQFlCQMC8lAWjqUMpTHrB/view?usp=sharing)
> * [Cummulative knockout (both forward and reverse knockout)](https://drive.google.com/file/d/1JI-N2oTszYtTH8mY_ZYG8op7r2PfiWWS/view?usp=sharing)
>
> The results reveal that **attention mass and causal importance are dissociated**:
>
> * Per-layer knockout at early layers degrades reconstruction while slightly improving downstream accuracy, indicating that early layers causally drive the downstream disruption.
> * Forward cumulative knockout rapidly restores downstream accuracy once the first several layers are masked
> * Reverse cumulative knockout does not recover downstream performance until masking reaches the early layers.
>
>
> This **forward/reverse asymmetry** is the core causal argument: downstream collapse
> is driven by early-layer interactions with the compression embedding. Late layers, despite
> exhibiting the highest attention mass, have minimal causal impact on downstream performance
> when knocked out  -  the observed concentration is a symptom rather than the cause.
>
>
>
>
>
>
> ---
>
>
>
>
>
> # Questions
>
> ### Q1. Limited number od samples
>
> See **W1**.
>
>
>
>
>
> ### Q2. Causality analysis for attention hijacking
>
> See **W2**.
>
>
>
>
>
> ### Q3. Downstream evaluation with progressive cramming.
>
> We present downstream evaluation results for progressive cramming in [Table 7](https://drive.google.com/file/d/1DIjiTpyIqQ4xuW0k9Q7mjCOFuXUPz2YX/view?usp=sharing). For progressive cramming, we report metrics computed exclusively over fully converged samples (i.e., those achieving perfect reconstruction). The performance drop relative to the baseline is comparable for both full and progressive cramming across all models, ranging from approximately 34%--38%.
>
> Note that Gemma-3-4B shows 0% convergence under progressive cramming, meaning no sample achieved perfect token-by-token reconstruction. Notably, full cramming still yields a relatively small performance drop for Gemma (54.97% vs. 57.07%). This finding requires further research.
>
>
>
>
> ---
>
>
>
>
> ## Erratum: Downstream evaluation bug
>
> We identified a bug in the perplexity computation for the HellaSwag and ARC benchmarks in the originally submitted version. Specifically, the compression embedding introduced an off-by-one token shift that was not accounted for during likelihood scoring, resulting in the near-random accuracy reported in the original paper. The corrected results are shown above.

---

> > ### Author Rebuttal · Reviewer_neRn · 2026-04-04
> >
> > Thank you for the detailed clarifications. Since my initial assessment was already positive, I will maintain my original score.

---

### Decision · Program_Chairs · 2026-04-30

**Decision:**

Accept (regular)

**Comment:**

I strongly agree with some of the reviewers’ comments that this paper has clear corrective value for the community. Although reviewers still expressed concerns about attention hijacking, I believe these issues can be addressed in the final version. I would encourage the authors to reorganize the paper somewhat and place greater emphasis on the major cramming issue.